# Randomized and Deterministic Maximin-share Approximations for Fractionally Subadditive Valuations

**Hannaneh Akrami**
Max Planck Institute for Informatics
Graduiertenschule Informatik
Universität des Saarlandes
`hakrami@mpi-inf.mpg.de`

**Kurt Mehlhorn**
Max Planck Institute for Informatics
Universität des Saarlandes
`mehlhorn@mpi-inf.mpg.de`

**Masoud Seddighin**
Tehran Institute for Advanced Studies
`m.seddighin@teias.institute`

**Golnoosh Shahkarami**
Max Planck Institute for Informatics
Graduiertenschule Informatik
Universität des Saarlandes
`gshahkar@mpi-inf.mpg.de`

## Abstract

We consider the problem of guaranteeing maximin-share (MMS) when allocating a set of indivisible items to a set of agents with fractionally subadditive (XOS) valuations. For XOS valuations, it has been previously shown that for some instances no allocation can guarantee a fraction better than $1/2$ of maximin-share to all the agents. Also, a deterministic allocation exists that guarantees $0.219225$ of the maximin-share of each agent. Our results involve both deterministic and randomized allocations. On the deterministic side, we improve the best approximation guarantee for fractionally subadditive valuations to $3/13 = 0.230769$. We develop new ideas on allocating large items in our allocation algorithm which might be of independent interest. Furthermore, we investigate randomized algorithms and the Best-of-both-worlds fairness guarantees. We propose a randomized allocation that is $1/4$-MMS ex-ante and $1/8$-MMS ex-post for XOS valuations. Moreover, we prove an upper bound of $3/4$ on the ex-ante guarantee for this class of valuations.

## 1 Introduction

Fair allocation has been a central problem in economics for decades. This problem arises naturally in real-world applications such as advertising, negotiation, rent sharing, inheritance, etc Caragiannis et al. (2019); Dehghani et al. (2018); Dickerson et al. (2014); Foley (1967); Nash Jr (1950); Varian (1973). In discrete fair division, the basic scenario is that we want to distribute a set $\mathcal{M}$ of $m$ indivisible items among $n$ agents, such that the allocation is deemed fair by the agents. Each agent $i$ has a valuation function $v_i : 2^{\mathcal{M}} \to \mathbb{R}^+$ that represents her happiness for receiving a subset of items.

How do we evaluate fairness? This question has been the subject of intense debates in various contexts, including philosophy, economics, distributive justice, and mathematics. Since the introduction of the cake-cutting[1] problem Steinhaus (1948), scientists have suggested several different notions to evaluate fairness. The challenge is that a fairness notion must be both reasonable in terms of justice and implementable in practice. Conceived by Steinhaus, proportionality is one of the most natural

---

[1] Cake-cutting is the continuous version of the fair allocation problem where the resource is a divisible cake.

37th Conference on Neural Information Processing Systems (NeurIPS 2023).

and prominent notions. An allocation is *proportional* if the share allocated to every agent is worth at least $1/n$ of her value for the entire resource.

Unfortunately, despite many positive results on proportionality for cake-cutting, simple examples indicate that proportionality is not a proper fairness criterion for the case of indivisible items. For example, when there is one item and two agents, one agent receives nothing, though her proportional share is non-zero assuming that the item has positive value for both of the agents. An alternative form of proportionality adopted to deal with indivisibilities is *Maximin-share* (MMS). For every agent $i$, the maximin-share of agent $i$, denoted by $\mathsf{MMS}_i$ is defined as follows:

$$\mathsf{MMS}_i = \max_{\langle \Pi_1, \Pi_2, \ldots, \Pi_n \rangle \in \Omega} \min_{1 \leq j \leq n} v_i(\Pi_j),$$

where $\Omega$ is the set of all partitions of $\mathcal{M}$ into $n$ parts. It is known that even when all the valuations are additive, MMS allocations, in which each agent $i$ gets at least $\mathsf{MMS}_i$, need not exist Feige et al. (2022); Kurokawa et al. (2018); Procaccia and Wang (2014). Nevertheless, several studies in recent years show that it is possible to guarantee a constant factor of maximin-share to all agents for various classes of valuation functions, including additive, submodular, and fractionally subadditive. See Table 1 for the state-of-the-art guarantees on the maximin-share. For Submodular valuation class, a 10/27-MMS allocation can be computed in polynomial time. For all other classes of valuation functions mentioned in Table 1, the states-of-the-art are existential results.

Table 1: A summary of the results for MMS in different valuation classes.

| Valuation Class | Approximation Guarantee | Upper bound |
|---|---|---|
| Additive | $\frac{3}{4} + \frac{1}{3836}$ Akrami and Garg (2024) | $\frac{39}{40}$ Feige et al. (2022) |
| Submodular | $\frac{10}{27}$ Uziahu and Feige (2023) | $\frac{3}{4}$ Ghodsi et al. (2018) |
| Fractionally Subadditive | 0.219225 Seddighin and Seddighin (2022) | $\frac{1}{2}$ Ghodsi et al. (2018) |
| Subadditive | $\frac{1}{\log n \log \log n}$ Seddighin and Seddighin (2022) | $\frac{1}{2}$ Ghodsi et al. (2018) |

Let us revisit the instance with one item and two agents. Suppose that the item has value 6 for both agents. By definition, the proportional share of each agent is $6/2 = 3$, and since one agent receives no item, satisfying proportionality or any approximation of it is impossible. On the other hand, we have $\mathsf{MMS}_1 = \mathsf{MMS}_2 = 0$. Thus, allocating the item to any agent satisfies maximin-share. Indeed, we can circumvent the non-guaranteed existence of fair allocation by reducing our expectation of fairness to maximin-share. However, regardless of how we allocate the item, one agent receives one item, and the other receives nothing. Therefore, having one agent with zero utility is inevitable for any deterministic allocation in this example. The question then arises: can we do better? One way to improve the allocation is to use randomization and obtain a better guarantee in expectation (*ex-ante*). For example, we can allocate the item to each agent with probability $1/2$. This way, the expected utility of each agent is equal to 3. In economic terms, this allocation satisfies proportionality *ex-ante*. Note that one agent receives no item *ex-post* (that is, after fixing the outcome); however, it guarantees proportionality ex-ante to both agents.

Considering random allocations and ex-ante fairness makes the problem much handier. For instance, assuming there are $n$ items and $n$ agents, allocating each item to each agent with probability $1/n$ satisfies proportionality ex-ante. However, this randomized allocation has no ex-post fairness guarantee: with a non-zero probability, the outcome allocates all the items to one agent, and the rest of the agents receive no item. It is tempting to find allocations that simultaneously admit ex-ante and ex-post guarantees. The support of such an allocation is limited to outcomes with some desirable fairness guarantee. For example, consider the following random allocation: we choose a random permutation of these $n$ items and allocate the $i^{\text{th}}$ item in the permutation to agent $i$. This allocation satisfies proportionality ex-ante and maximin-share ex-post.

Recently, several studies have investigated randomized allocations with both ex-ante and ex-post guarantees. Some notable results with the focus on additive valuations are (i) an ex-ante envy-free and ex-post EF1 allocation algorithm Freeman et al. (2020), (ii) an ex-ante proportional and ex-post 1/2-MMS allocation algorithm Babaioff et al. (2022), and (iii) an ex-post 3/4-MMS and ex-ante 0.785-MMS allocation algorithm Akrami et al. (2023b). Very recently, Feldman et al. (2023) studied best-of-both-worlds for subadditive valuations and gave an allocation algorithm with ex-ante guarantee of 1/2-envy-freeness and ex-post guarantee of 1/2-EFX and EF1.

In this paper, our goal is to explore fair deterministic and randomized allocations for fractionally subadditive valuation functions. A valuation function $v_i(\cdot)$ is fractionally subadditive (XOS), if there exists a set of several additive valuation functions $u_{i,1}, u_{i,2}, \ldots, u_{i,\ell} : 2^{\mathcal{M}} \to \mathbb{R}_{\geq 0}$ such that for every set $S$ we have $v_i(S) = \max_{1 \leq k \leq \ell} u_{i,k}(S)$. Fractionally subadditive is a super-class of different set functions such as additive, gross substitute, and submodular. In addition, fractionally subadditive is a subclass of subadditive set functions.

The fairness notion we consider is maximin-share. Indeed, we are looking for allocations that satisfy an approximation of maximin-share both ex-ante and ex-post for fractionally subadditive valuations. Of course, we expect our ex-ante guarantee to be stronger than the ex-post one. Moreover, we improve the ex-post guarantee (deterministically) for this class of valuations. We refer to the next subsection for an overview on our results and techniques.

## 1.1 Our Results and Techniques

In this paper, we provide improved guarantees for maximin-share in the fractionally subadditive setting. We investigate both randomized and deterministic allocation algorithms.

**Randomized Allocations**    For randomized allocations, we take one step toward extending the best-of-both-worlds idea for fairness concepts to valuations more general than additive. For the additive setting, Babaioff et al. (2022) proved the existence of randomized allocations that are proportional ex-ante and $1/2$-MMS ex-post. However, as we show in Section 3, though guaranteeing proportionality ex-ante is easy for XOS valuations, this notion is not always a proper choice as a fairness criterion. In fact, for some instances, proportionality can be as small as $O(1/n)$ of the MMS value, which is highly undesirable. Therefore, here we focus on guaranteeing MMS both ex-ante and ex-post. In contrast to the additive setting, for XOS valuations guaranteeing MMS ex-ante is not easy. Recall that in the additive setting, a fractional allocation that allocates a fraction $1/n$ of each item to each agent is proportional and consequently MMS. However, for some XOS instances this allocation is $O(1/n)$-MMS (Observation 3.2). Indeed, we show that there are instances for which guaranteeing MMS ex-ante is not possible. More precisely, we show that there are instances that no randomized allocation can guarantee a factor better than $3/4$ of maximin-share to all agents (Lemma 3.3). On the positive side, we propose an algorithm that finds a randomized allocation which is $1/4$-MMS ex-ante and $1/8$-MMS ex-post (Theorem 1.1). The idea to prove the ex-ante approximation guarantee of our allocation is inspired by Ghodsi et al. (2018). In the beginning of the algorithm, as long as there exists a remaining agent $i$ who likes a remaining item $b_j$ at least $1/2$ of her MMS value, we allocate $b_j$ to $i$ and remove them from $\mathcal{M}$ and $\mathcal{N}$. By the end of this phase, we get an upper bound on the value of the remaining items for the remaining agents. Let $\bar{v}_i(\cdot) = \min\{\frac{1}{2}, v_i(\cdot)\}$ for all agents $i$. In the second phase, we find a half-integral allocation that maximizes social welfare with respect to these new valuations $\bar{v}_i$. For every $i$, $\bar{v}_i(\cdot)$ is the same as $v_i(\cdot)$, except that for every bundle $X$ with $v_i(X) > 1/2$ we have $\bar{v}_i(X) = 1/2$. Then, we convert this fractional allocation into a randomized one. To prove the ex-post guarantee, the fact that the allocation is half-integral plays a key role.

**Theorem 1.1.** *There exists a randomized allocation that is $1/4$-MMS ex-ante and $1/8$-MMS ex-post.*

**Deterministic Allocations**    Since the impossibility result on MMS by Procaccia and Wang (2014), there has been a considerable effort in the line of improving the approximate MMS factor for various classes of the valuation functions Akrami and Garg (2024); Akrami et al. (2023a); Barman and Krishnamurthy (2020); Garg and Taki (2021); Ghodsi et al. (2018); Kurokawa et al. (2018). However, all these results and their techniques apply to less general classes of valuation functions than XOS. In this work, we give a deterministic algorithm which outputs an allocation with $3/13$-MMS guarantee when valuation functions are XOS. This way, we improve the state-of-the-art on approximate MMS for XOS valuations from $0.219225$ Seddighin and Seddighin (2022) to $3/13 = 0.230769$. In order to do so, like in our randomized algorithm, we first allocate large items. However, in addition to allocating single large items, here we also allocate pairs and triples of items if they satisfy some agent up to $3/13$ factor of their MMS value. This way we get a stronger upper bound on the value of most of the remaining items. In the last step, we output an allocation which maximizes the social welfare with respect to valuations $\bar{v}_i(\cdot) = \min\{\frac{6}{13}, v_i(\cdot)\}$ for the remaining agents and items. Note that this last step is also in correspondence with the last step of the randomized algorithm. Here we cap the value of the bundles with $6/13$ (instead of $1/2$) and we output an integral allocation with maximum

social welfare (instead of a half-integral allocation with maximum social welfare). Moreover, here we need to do a more careful analysis to show that the output is indeed a $3/13$-MMS allocation.

**Theorem 1.2.** *Given an instance with* XOS *valuations, there exists a* $3/13$-MMS *allocation.*

Both of our results are constructive; however, their running times are not polynomially bounded. One step that cannot be done in polynomial time is computing the MMS values of the agents and another one is computing an allocation with maximum social welfare.

### 1.2 Further Related Work

**Maximin-share fairness notion.** Budish (2011) introduced the MMS notion. As we already mentioned, there are instances with additive valuations for which no MMS allocation exists Feige et al. (2022); Kurokawa et al. (2018); Procaccia and Wang (2014). Therefore, several studies considered the approximation guarantees. For additive valuations, the first approximation guarantee, $2/3$-MMS, is given by Kurokawa et al. (2018). The approximation factor improved over time by Ghodsi et al. (2018) to $3/4$, and then to $3/4 + o(1)$ by Garg and Taki (2021) and Akrami et al. (2023a). The best-known result for this valuation classed is $3/4 + 3/3836$ which was very recently proposed by Akrami and Garg (2024). Moreover, there are several works on MMS approximations for submodular valuations Barman and Krishnamurthy (2020); Ghodsi et al. (2018), and subadditive valuations Ghodsi et al. (2018); Seddighin and Seddighin (2022). Also, Farhadi et al. (2019) proved the tight bound of $1/n$-MMS for the setting where agents can have different entitlements.

**Fractionally subadditive valuations.** A $1/5$-MMS approximation algorithm for fractionally subadditive valuations proposed by Ghodsi et al. (2018), and improved later to $0.2173913$-MMS approximation by Seddighin and Seddighin (2022). Fractionally subadditive valuations for hereditary set systems have been studied by Li and Vetta (2018). Different other notions of fairness have been studied for this class of valuation. Feige (2009) studied maximizing social welfare. Also, Hoefer et al. (2023) proved the existence of randomized allocations which are proportional ex-ante and proportional up to one item (PROP1) ex-post.

**Best-of-both-worlds fairness.** Aleksandrov et al. (2015) first explored this line of research in the context of the food bank problem for a special case of additive valuations. Freeman et al. (2020) proposed a randomized polynomial-time algorithm for additive valuations that is envy-free ex-ante and EF1 ex-post. Afterward, Aziz (2020) modified this algorithm to also get the Probabilistic Serial fractional outcome Bogomolnaia and Moulin (2001) as well as a weak notion of efficiency with a simpler proof. Babaioff et al. (2022) provided an allocation algorithm in which the expected value of each agent's bundle is at least a 1/n-fraction of her value for the set of all items (ex-ante proportional) and ex-post proportional up to one item and $1/2$-MMS allocations for the additive valuations. Feldman et al. (2023) gave an allocation algorithm for subadditive valuations with guarantees of $1/2$-envy-freeness ex-ante and $1/2$-EFX and EF1 ex-post. Moreover, studies have been conducted for additive valuations with binary marginals Aziz (2020); Halpern et al. (2020), and for matroid rank valuations Aziz et al. (2023); Babaioff et al. (2021).

## 2   Preliminaries

A fair division instance is denoted by $\mathcal{I} = (\mathcal{N}, \mathcal{M}, \mathcal{V})$ where $\mathcal{N} = [n]$ is a set of $n$ agents, $\mathcal{M}$ is a set of $m$ indivisible items and $\mathcal{V} = (v_1, \ldots, v_n)$ is a vector of valuation functions. Each agent $i$ has a valuation function $v_i : 2^{\mathcal{M}} \to \mathbb{R}_{\geq 0}$ that represents her value for every bundle of items. Thus, for every set $S$ of items, $v_i(S)$ shows the value of $S$ to agent $i$. We denote the $j^{\text{th}}$ item by $b_j$. An allocation of items is a partition of $\mathcal{M}$ into $n$ parts (i.e., bundles) where the $i^{\text{th}}$ bundle is the share allocated to agent $i$. For an allocation $\mathcal{A}$, we denote the bundle of agent $i$ by $\mathcal{A}_i$. Here we assume that the valuations are monotone, i.e., for every two sets $S$ and $T$ such that $S \subseteq T$, we have $v_i(S) \leq v_i(T)$, and normalized, i.e., $v_i(\emptyset) = 0$ for all $i \in \mathcal{N}$. Our discussion in this paper involves two classes of valuation functions: additive and fractionally subadditive valuation functions. A valuation function $v_i(\cdot)$ is additive, if for every set $S$ of items, we have $v_i(S) = \sum_{b_j \in S} v_i(\{b_j\})$.

**Definition 2.1** (XOS)**.** A valuation function $v_i(\cdot)$ is fractionally subadditive (XOS), if there exists a set of several additive valuation functions $u_{i,1}, u_{i,2}, \ldots, u_{i,\ell} : 2^{\mathcal{M}} \to \mathbb{R}_{\geq 0}$ such that for every set $S$ we have $v_i(S) = \max_{1 \leq k \leq \ell} u_{i,k}(S)$.

Given an allocation $\mathcal{A}$, we denote by $u_{i,i'}$, an additive function of $v_i$ that defines $v_i(\mathcal{A}_i)$, i.e., $v_i(\mathcal{A}_i) = u_{i,i'}(\mathcal{A}_i)$. Another term we frequently use in this paper is contribution; which is defined to evaluate the marginal value of one set to another.

**Definition 2.2** (Contribution). For every sets $S, T$ of items such that $S \subseteq T$, we define the marginal contribution of $S$ to $T$ with respect to valuation function $v$, denoted by $C_v^T(S)$ as $C_v^T(S) = v(T) - v(T \setminus S)$, i.e., the contribution of $S$ to $T$ is the decrease in the value when $S$ is removed from $T$.

With abuse of notation, for an allocation $\mathcal{A}$ of items to agents with valuation vector $\mathcal{V} = (v_1, \ldots, v_n)$ and every set $S$ of items, we define the contribution of $S$ to $\mathcal{A}$ with respect to $\mathcal{V}$, denoted by $C_\mathcal{V}^\mathcal{A}(S)$ as $C_\mathcal{V}^\mathcal{A}(S) = \sum_{1 \leq i \leq n} C_{v_i}^{\mathcal{A}_i}(\mathcal{A}_i \cap S)$. However, since an XOS valuation function might include many additive functions, this equation is not always practical. Therefore, we bound $C_\mathcal{V}^\mathcal{A}(S)$ as follows.

**Observation 2.3.** Given an allocation $\mathcal{A}$ of items to agents with valuation vector $\mathcal{V}$, and every set $S$ of items, we have $C_\mathcal{V}^\mathcal{A}(S) \leq \sum_{1 \leq i \leq n} u_{i,i'}(\mathcal{A}_i \cap S)$.

Our goal is to allocate the items to the agents in a fair manner. Here we discuss two share-based notions of fairness, namely proportionality, and maximin-share. For agent $i$, we define the proportional share of agent $i$, denoted by $\pi_i$ as $\pi_i = v_i(\mathcal{M})/n$. We also define the maximin-share of agent $i$, denoted by $\mathsf{MMS}_i$ as

$$\mathsf{MMS}_i = \max_{\langle \Pi_1, \Pi_2, \ldots, \Pi_n \rangle \in \Omega} \min_{1 \leq j \leq n} v_i(\Pi_j),$$

where $\Omega$ is the set of all partitions of $\mathcal{M}$ into $n$ bundles. Moreover, if for a partition $\Pi = \langle \Pi_1, \Pi_2, \ldots, \Pi_n \rangle$ of $\mathcal{M}$ into $n$ bundles we have $v_i(\Pi_j) \geq \mathsf{MMS}_i$ for all $j \in [n]$, we say $\Pi$ is an "MMS partition" of agent $i$. For brevity, in the rest of the paper, we assume that the valuations are scaled so that for each agent $i$, we have $\mathsf{MMS}_i = 1$. We also define approximate versions of these two notions as follows. For a constant $\alpha > 0$, we say an allocation is $\alpha$-proportional, if it guarantees to each agent $i$ a bundle with value at least $\alpha \pi_i$. Likewise, in an $\alpha$-MMS allocation, the value of the share allocated to each agent $i$ is at least $\alpha$.

**Randomized allocation.** In this paper, we also consider randomized allocations. A randomized allocation is a distribution over a set of deterministic allocations. For a randomized allocation $\mathcal{R}$, we denote by $D(\mathcal{R})$ the set of allocations in the support of $\mathcal{R}$. Given a randomized allocation $\mathcal{R}$, the expected welfare of agent $i$ for $\mathcal{R}$ is defined as $v_i(\mathcal{R}) = \sum_{\mathcal{A} \in D(\mathcal{R})} v_i(\mathcal{A}_i) \cdot p_\mathcal{A}$, where $p_\mathcal{A}$ is the probability of allocation $\mathcal{A}$ in $\mathcal{R}$.

**Fractional allocation.** En route to proving our results, we leverage another relaxed form of allocation called fractional allocation. In a fractional allocation, we ignore the indivisibility assumption and treat each item as a divisible one. Formally, a fractional allocation $\mathcal{F}$ is a set of $nm$ variables $f_{i,j}$ indicating the fraction of item $b_j$ allocated to agent $i$. Therefore, we expect allocation $\mathcal{F}$ to satisfy the following constraints:

$$\forall_j \qquad \sum_{1 \leq i \leq n} f_{i,j} \leq 1 \qquad\qquad \text{(we have one unit of each item)}$$

$$\forall_{i,j} \qquad 0 \leq f_{i,j} \qquad \text{(each agent receives a non-negative share of each item)}$$

A fractional allocation is *complete* if $\sum_i f_{i,j} = 1$ for all items $b_j$, i.e., all items are completely allocated. Given a fractional allocation $\mathcal{F}$, we define the utility of agent $i$ for $\mathcal{F}$ in the same way as we calculate it for integral allocations: $v_i(\mathcal{F}) = \max_k \sum_{1 \leq j \leq m} u_{i,k}(\{b_j\}) f_{i,j}$. Complete fractional allocations give rise to randomized allocations in the standard way, i.e., the probability $p_\mathcal{A}$ of an allocation $\mathcal{A}$ is defined as

$$p_\mathcal{A} = \prod_{1 \leq i \leq n} \prod_{b_j \in \mathcal{A}_i} f_{ij}. \tag{1}$$

**Lemma 2.4.** *Let $\mathcal{F}$ be a complete fractional allocation and let randomized allocation $\mathcal{R}$ be defined by (1). Then given an XOS valuation function $v_i$, $v_i(\mathcal{R}) \geq v_i(\mathcal{F})$ for all $1 \leq i \leq n$.*

We also need to redefine *contribution* for fractional allocations and fractional bundles. Suppose that $\mathcal{F}$ is a fractional allocation and $S$ is a fractional set of items. Since items are fractionally allocated, the term contribution must be defined more precisely. For example, suppose that set $S$ consists of a

fraction $0.4$ of item $b_j$, and in allocation $\mathcal{F}$, $0.2$ of $b_j$ belongs to agent $i_1$, $0.5$ of $b_j$ belongs to agent $i_2$, and $0.3$ of $b_j$ belongs to agent $i_3$. Here we should exactly determine that this $0.4$ fraction of item $b_j$ in $S$ corresponds to the share of which agent or agents. One reasonable strategy is to choose the share of each agent in a way that after removal of $S$ from $\mathcal{F}$ we have the smallest possible decrease in the social welfare. Based on this strategy, assuming that $s_j$ is the fraction of item $b_j$ in $S$, we define the contribution of $S$ to $\mathcal{F}$, denoted by $C_{\mathcal{V}}^{\mathcal{F}}(S)$ as the answer of the following optimization program:

$$\text{minimize} \quad \sum_{1 \leq i \leq n} v_i(\mathcal{F}) - v_i(\mathcal{F}')$$

$$\text{subject to} \quad \sum_{1 \leq i \leq n} f_{i,j} - f'_{i,j} = s_j \qquad\qquad \forall_j$$

$$0 \leq f'_{i,j} \leq f_{i,j} \qquad\qquad \forall_{i,j} \qquad (2)$$

Generally, it is hard to deal with the above optimization program. Here, we use an important property of $C_{\mathcal{V}}^{\mathcal{F}}(\cdot)$ to obtain our results.

**Lemma 2.5.** *Let $\mathcal{F}$ be an arbitrary fractional allocation and assume that for every agent $i$, $v_i(\cdot)$ is* XOS*. Then, for every partition of the items into fractional sets $S_1, S_2, \ldots, S_t$, we have $\sum_{1 \leq k \leq t} C_{\mathcal{V}}^{\mathcal{F}}(S_k) \leq \sum_{1 \leq i \leq n} v_i(\mathcal{F})$.*

For a randomized allocation, we define two types of fairness guarantees, namely *ex-ante* and *ex-post* as in Definitions 2.6 and 2.7.

**Definition 2.6** (ex-ante)**.** Given a randomized allocation $\mathcal{R}$, we say $\mathcal{R}$ is $\alpha$-MMS *ex-ante*, if for every agent $i$, we have $v_i(\mathcal{R}) \geq \alpha$. Similarly, $\mathcal{R}$ is $\alpha$-proportional, if for every agent $i$, $v_i(\mathcal{R}) \geq \alpha \pi_i$.

**Definition 2.7** (ex-post)**.** An allocation $\mathcal{R}$ is $\alpha$-MMS *ex-post*, if for every allocation $\mathcal{A} \in D(\mathcal{R})$, we have that allocation $\mathcal{A}$ is $\alpha$-MMS. Also, we say $\mathcal{R}$ is $\alpha$-proportional ex-post if every allocation in the support of $\mathcal{R}$ is $\alpha$-proportional.

One tool that we refer to in this paper is the result of Babaioff et al. (2022) for converting a fractional allocation into a faithful randomized allocation.

**Theorem 2.8** (Proved by Babaioff et al. (2022))**.** *Assume that the valuations are additive and let $\mathcal{F}$ be a fractional allocation. Then there exists a randomized allocation $\mathcal{R}$ such that the ex-ante utility of the agents for $\mathcal{R}$ is the same as the utility of the agents in $\mathcal{F}$, and for every allocation $\mathcal{A}$ in the support of $\mathcal{R}$ the following holds: $\forall_i, v_i(\mathcal{A}_i) \geq v_i(\mathcal{R}) - \max_{j:f_{i,j} \notin \{0,1\}} v_i(\{b_j\})$.*

In this paper, we use a more delicate analysis of the method used in Theorem 2.8 to convert fractional allocations into randomized ones. This helps us improve our ex-post approximation guarantees.

## 3 Ex-ante Guarantees

In this section, our goal is to explore the possibility of designing randomized allocation that is $\alpha$-MMS ex-ante or $\alpha$-proportional ex-ante. Note that, in contrast to the additive case, for XOS valuations there is no meaningful correspondence between proportionality and maximin-share; proportional share can be larger or smaller than maximin-share. Recall that for the additive case, we always have $\pi_i \geq \mathsf{MMS}_i$ and therefore, maximin-share is implied by proportionality. For fractionally subadditive valuations, however, $\pi_i$ can be as small as $\mathsf{MMS}_i/n$.

For the additive setting, a simple fractional allocation that allocates a fraction $1/n$ of each item to each agent guarantees proportionality and consequently maximin-share. Using Theorem 2.8 one can convert this allocation to a randomized allocation that is proportional ex-ante. In Observation 3.1 we show that proportionality can be guaranteed ex-ante for XOS valuations.

**Observation 3.1.** Every randomized allocation that allocates each item with probability $1/n$ to each agent is proportional ex-ante.

In contrast to the additive setting, finding an allocation that guarantees maximin-share is not trivial. Indeed, the simple fractional allocation that guarantees proportionality in Observation 3.1 can be as bad as $O(1/n)$-MMS.

**Observation 3.2.** Given any instance $\mathcal{I}$, let $\mathcal{F}(\mathcal{I})$ be the fractional allocation that allocates a fraction $1/n$ of each item to each agent. Then, there exists an instance $\mathcal{I}$ such that the maximin-share guarantee of $\mathcal{F}(\mathcal{I})$ is $O(1/n)$.

Generally, there are two main challenges in the process of designing a randomized allocation that guarantees an approximation of maximin-share. In contrast to the additive setting, finding a fractional or randomized allocation that approximates maximin-share is not easy. As well as that, transforming a fractional allocation into a randomized one is not straightforward. Indeed, as we show in Lemma 3.3, neither fractional allocations nor randomized allocations can guarantee MMS. We prove an upper bound on the best approximation guarantee of each one of these allocation types.

**Lemma 3.3.** *For* XOS *valuations, the best* MMS *guarantee for fractional allocations and the best ex-ante* MMS *guarantee for randomized allocation is upper bounded by* $3/4$.

Before we prove our lower bound on the maximin-share guarantee for randomized allocations, we note that another challenge about XOS valuations is that in sharp contrast to additive valuations, transforming a fractional allocation to a randomized one is not easy. Indeed, we can show that for a fractional allocation $\mathcal{F}$ there might be randomized allocations $\mathcal{R}$ and $\mathcal{R}'$ with different utility guarantees for the agents, such that in both $\mathcal{R}$ and $\mathcal{R}'$ the probability that each item $b_j$ is allocated to agent $i$ is equal to $f_{i,j}$. Example 3.4 gives more insight into this challenge.

**Example 3.4.** Consider the following instance: there are $n^2$ items. The valuation of agent $i$ is an XOS set function consisting of $n$ additive valuation functions as follows: partition the items into $n$ bundles each with $n$ items. For each additive function $u_{i,k}$, the value of each item in the $k^{\text{th}}$ bundle is $1/n$ and the value of the rest of the items is 0. Define allocations $\mathcal{R}$ and $\mathcal{R}'$ as follows: $(i)$: allocation $\mathcal{R}$ allocates each item to each agent with probability $1/n$, and $(ii)$: allocation $\mathcal{R}'$ considers a random permutation of the bundles in the optimal MMS partition of the agents and allocates the $i^{\text{th}}$ bundle in the permutation to agent $i$. In both of these allocations, each item is allocated to each agent with probability $1/n$. However, the maximin-share guarantee of $\mathcal{R}$ is $O(\log n/n)$. On the other hand, allocation $\mathcal{R}'$ guarantees value 1 to all agents.

Despite these hurdles, in Theorem 3.5 we show that there exists a randomized allocation that guarantees $1/4$-MMS to all the agents ex-ante. To prove Theorem 3.5, we first show that a fractional allocation exists that is $1/4$-MMS. Next, we convert it to a randomized allocation. Theorem 3.5 along with Lemma 3.3 leave a gap of $[1/4, 3/4]$ between the best upper bound and the best lower-bound for the maximin-share guarantee of fractional allocations in the XOS setting.

**Theorem 3.5.** *There exists a randomized allocation that is* $1/4$-MMS *ex-ante.*

# 4 Ex-ante and Ex-post Guarantees

Unfortunately, the randomized allocation obtained by Theorem 3.5 has no ex-post fairness guarantee. The issue is that we use Theorem 2.8 to convert the fractional allocation into a randomized one. However, Theorem 2.8 only guarantees that the ex-post value of each agent is at least the value of her fractional allocation minus the value of the heaviest item which is partially (and not fully) allocated to her in the fractional allocation. However, currently, we have no upper bound on the value of the allocated items, and therefore, the ex-post value of an agent might be close to 0. To resolve this, we allocate valuable items beforehand to keep the value of the remaining items as small as possible. In Lemma 4.1 we explain a simple and very practical fact that is frequently used in the previous studies. Ghodsi et al. (2018); Seddighin and Seddighin (2022); Barman and Krishna Murthy (2017); Amanatidis et al. (2019).

**Lemma 4.1.** *Removing one item and one agent from the instance does not decrease the maximin-share value of the remaining agents for the remaining items.*

Given that our goal is to construct a randomized allocation which is $1/4$-MMS ex-ante, by Lemma 4.1 we can assume without loss of generality that the value of each item to each agent is less than $1/4$; otherwise, we can reduce the problem using Lemma 4.1. However, a combination of this assumption and Theorem 3.5 still gives no ex-post guarantee. To improve the ex-post guarantee, we revisit the proof of Theorem 2.8 and show that for our setting, a stronger guarantee can be achieved using the matching method for converting a fractional allocation into a randomized one.

**Lemma 4.2.** *Assume that the valuations are additive and let $\mathcal{F}$ be a complete fractional allocation with $f_{ij} \in \{0, 1/2, 1\}$ for all $i$ and $j$. Then there is a randomized allocation $\mathcal{R}$ with $D(\mathcal{R}) = \{\mathcal{A}^1, \mathcal{A}^2\}$, such that*

- *For every agent $i$ we have $v_i(\mathcal{R}) = v_i(\mathcal{F})$.*

- *For every agent $i$ we have*

$$\min\left\{v_i(\mathcal{A}_i^1), v_i(\mathcal{A}_i^2)\right\} \geq v_i(\mathcal{R}) - \frac{\max\{v_i(b_j) \mid f_{ij} = 1/2\}}{2}.$$

In Lemma 4.2, we show that we can find a fractional allocation with a special structure that makes the transformation step more efficient. These ideas together help us achieve a randomized allocation with $1/4$-MMS guarantee ex-ante and $1/8$-MMS guarantee ex-post. In order to do so, we use the following algorithm:

1. While there exists an item $b_j$ with value at least $1/4$ to an agent $i$, allocate $b_j$ to agent $i$ and remove $i$ and $b_j$ respectively from $\mathcal{N}$ and $\mathcal{M}$.

2. For the remaining agents $[n]$ and items proceed as follows: for every agent $i$, define $\bar{v}_i$ as follows: for every subset $S$ of items, $\bar{v}_i(S) = \min(t, v_i(S))$. Let $\bar{v} = (\bar{v}_1, \dots, \bar{v}_n)$ and return a half-integral allocation $\mathcal{F}$ that maximizes the social welfare with respect to $\bar{v}$, i.e., $\mathcal{F} = \arg\max_{F \in \Pi} \sum_{i \in \mathcal{N}'} \bar{v}_i(F_i)$ where $\Pi$ is the set of all half-integral allocations of the remaining items to the remaining agents.

3. Convert $\mathcal{F}$ into a randomized allocation using Lemma 4.2.

The goal in Step 2 is to find a fractional allocation that is $1/4$-MMS. However, we want this allocation to have a special structure that facilitates constructing the randomized allocation. Therefore, instead of directly choosing the allocation that maximizes social welfare, we consider $\bar{v}_i$ as the valuation function of agent $i$ and return a half-integral allocation $\mathcal{F}$.

**Lemma 4.3.** $\mathcal{F}$ *is* $1/4$-MMS.

Now we are ready to prove Theorem 1.1.

**Theorem 1.1.** *There exists a randomized allocation that is* $1/4$-MMS *ex-ante and* $1/8$-MMS *ex-post.*

*Proof.* The ex-ante guarantee follows from Lemmas 4.3 and 1.

Let $\mathcal{A}^1$ and $\mathcal{A}^2$ be the integral allocations obtained by Lemma 4.2. Consider any agent $i$, and let $u_{i,i'}$ be such that $v_i(\mathcal{R}) = \sum_j f_{ij} u_{i,i'}(b_j)$. Then, by Lemma 4.2 for $r \in \{1, 2\}$ we have

$$u_{i,i'}(\mathcal{A}_i^r) \geq u_{i,i'}(\mathcal{R}_i^\ell) - \frac{\max\{u_{i,i'}(b_j) \mid f_{i,j} = 1/2\}}{2}$$

and since by Lemma 4.1 we know the value of each item for each agent is less than $1/4$, we have

$$v_i(\mathcal{A}_i^\ell) \geq v_i(\mathcal{R}_i) - \frac{\max_j v_i(b_j)}{2} > \frac{1}{4} - \frac{1}{8} = \frac{1}{8}.$$

Hence, the ex-post guarantee holds as well. $\qquad \square$

# 5 3/13-MMS Allocation

In this section, our goal is to improve the approximation guarantee of MMS for deterministic allocations in the fractionally subadditive setting. We show that guaranteeing a factor $3/13 \approx 0.230769$ of maximin-share to all the agents is always possible. Before this work, the best approximation guarantee for maximin-share in the XOS setting was $0.219225$-MMS Seddighin and Seddighin (2022).

Our algorithm for improving the ex-post guarantee is based on our previous algorithms plus two additional steps and a more in-depth analysis. In this algorithm, before finding the allocation that maximizes social welfare, we strengthen our upper bound on the value of items. For this, we add two more steps to our algorithm in which we satisfy some of the agents with two items and three items. In contrast to the first step (i.e., allocating single items to agents), these steps might decrease the maximin-share value of the remaining agents for the remaining items. Let $t = 6/13$. The goal is to find a $t/2$-MMS allocation. Our allocation algorithm is as follows:

1. If there exists an item $b_j$ with value at least $t/2$ to an agent $i$, allocate $b_j$ to agent $i$ and remove $i$ and $b_j$ respectively from $\mathcal{N}$ and $\mathcal{M}$.

2. If there exists a pair of items $b_j, b_k$ with the total value of at least $t/2$ to some agent $i$, allocate $\{b_j, b_k\}$ to agent $i$ and remove $i$ and $\{b_j, b_k\}$ from $\mathcal{N}$ and $\mathcal{M}$ respectively.

3. If there exists a triple of items $b_j, b_k, b_s$ with the total value of at least $t/2$ to some agent $i$, allocate $\{b_j, b_k, b_s\}$ to agent $i$ and remove $i$ and $\{b_j, b_k, b_s\}$ from $\mathcal{N}$ and $\mathcal{M}$ respectively.

4. For the remaining agents $\mathcal{N}'$ and items $\mathcal{M}'$, proceed as follows: for every agent $i$, define $\bar{v}_i$ as follows: for every subset $S$ of items, $\bar{v}_i(S) = \min(t, v_i(S))$. Let $\bar{v} = (\bar{v}_1, \ldots, \bar{v}_n)$ and return an allocation $\mathcal{A}$ that maximizes the social welfare with respect to $\bar{v}$, i.e., $\mathcal{A} = \arg\max_{A \in \Pi} \sum_{i \in \mathcal{N}'} \bar{v}_i(A_i)$ where $\Pi$ is the set of all allocations of $\mathcal{M}'$ to $\mathcal{N}'$.

In the rest of this section, we analyze the above algorithm. By Lemma 4.1, after Step 1, the MMS value of all the agents is at least 1. Let $n$ be the number of remaining agents after Step 1. We denote by $n_1$ and $n_2$, the number of agents that are satisfied in Steps 2 and 3 respectively and let $n' = n - n_1 - n_2 = |\mathcal{N}'|$ be the number of remaining agents after Step 3. In contrast to the first step, Step 2 and 3 might decrease the maximin-share value of the remaining agents for the remaining items. However, we prove that the remaining items satisfy special structural properties.

**Observation 5.1.** Since no item can satisfy any remaining agent after Step 1, for every agent $i$ and every item $b_j$, we have $v_i(\{b_j\}) < t/2$.

Also, by the method that we allocate the items in Step 3, after this step Observation 5.2 holds.

**Observation 5.2.** Since after Step 3, no triple of items can satisfy an agent, for every different items $b_j, b_k, b_s$ and every agent $i$ we have $v_i(\{b_j, b_k, b_s\}) < t/2$.

Note that since the valuations are XOS, Observation 5.2 implies no upper bound better than $t/2$ on the value of a single item to an agent. For example, consider the following extreme scenario: for a small constant $\epsilon > 0$, the value of every subset of items to agent $i$ is equal to $t/2 - \epsilon$. It is easy to check that this valuation function is XOS. For this case, the value of every triple of items is also equal to $t/2 - \epsilon$, but this implies no upper bound better than $t/2$ on the value of a single item.

**Lemma 5.3.** *Fix a remaining agent $i$ and consider the $n$ bundles with value at least 1 in an* MMS *partition of agent $i$ after Step 1. Put these bundles into 4 different sets $B_0, B_1, B_2, B_{\geq 3}$, where for $0 \leq \ell \leq 2$, set $B_\ell$ contains bundles that lose exactly $\ell$ items in Steps 2 and 3, and $B_{\geq 3}$ contains bundles that lose more than 4 items in these steps. After Step 3, the following inequality holds: $n' \leq |B_0| + \frac{2}{3}|B_1| + \frac{1}{3}|B_2|$.*

Finally, in Step 4, we find the integral allocation $\mathcal{A}$ that maximizes social welfare with respect to $\bar{v}$ for the remaining agents. Let $Z = \sum_{i \in \mathcal{N}'} \bar{v}_i(\mathcal{A}_i)$. Since for each remaining agent $i$, $\bar{v}_i(\mathcal{A}_i)$ is upper-bounded by $t$, we have $Z \leq n't$. If for every agent $i$, $v_i(\mathcal{A}_i) \geq t/2$ holds, then $\mathcal{A}$ is $t/2$-MMS, and we are done. Therefore, for the rest of this section, assume that for an agent $i^*$, we have $v_{i^*}(\mathcal{A}_{i^*}) < t/2$.

Let $B_0, B_1$, and $B_2$ be the sets defined for agent $i^*$ in Lemma 5.3. In Lemma 5.4, we give lower bounds on the contribution of the bundles in $B_0$, $B_1$ and $B_2$ to $\mathcal{A}$ respectively.

**Lemma 5.4.** *After Step 3, for all bundles*

- $X \in B_0$, *there exists a partition of $X$ into $X^1$ and $X^2$ such that $C_{\bar{v}}^{\mathcal{A}}(X^1) + C_{\bar{v}}^{\mathcal{A}}(X^2) \geq t$.*

- $X \in B_1$, *there exists a partition of $X$ into $X^1$ and $X^2$ such that $C_{\bar{v}}^{\mathcal{A}}(X^1) + C_{\bar{v}}^{\mathcal{A}}(X^2) \geq \frac{2}{3}t$.*

- $X \in B_2$, $C_{\bar{v}}^{\mathcal{A}}(X) \geq \frac{1}{2}t$.

**Theorem 1.2.** *Given an instance with* XOS *valuations, there exists a $3/13$-MMS allocation.*

*Proof.* Let $\mathcal{A}$ be the output of our Algorithm. Towards a contradiction, assume for agent $i^*$, $v_{i^*}(\mathcal{A}_{i^*}) < 3/13 = t/2$. For all agents $i$ which are removed during the first three steps, we have $v_i(\mathcal{A}_i) \geq t/2 = 3/13$. Therefore, $i^* \in \mathcal{N}'$. For all $X \in B_0$, let $X^1$ and $X^2$ be as defined in

Lemma 5.4. We have

$$t(n' - \frac{1}{2}) > \sum_{i \in \mathcal{N}'} \bar{v}_i(\mathcal{A}_i) \qquad \text{(for all } i \in \mathcal{N}', \bar{v}_i(\mathcal{A}_i) \leq t \text{ and } \bar{v}_{i^*}(\mathcal{A}_{i^*}) < t/2)$$

$$\geq \sum_{X \in B_0} \left( C_{\bar{v}}^{\mathcal{A}}(X^1) + C_{\bar{v}}^{\mathcal{A}}(X^2) \right) + \sum_{X \in B_1} \left( C_{\bar{v}}^{\mathcal{A}}(X^1) + C_{\bar{v}}^{\mathcal{A}}(X^2) \right) + \sum_{X \in B_2} C_{\bar{v}}^{\mathcal{A}}(X)$$

$$\text{(Lemma 2.5)}$$

$$\geq t|B_0| + \frac{2}{3}t|B_1| + \frac{1}{2}t|B_2| \geq tn'. \qquad \text{(Lemma 5.4 and Lemma 5.3)}$$

$$\square$$

# 6 Conclusion

In this paper, we developed randomized and deterministic allocations that guarantee approximations of maximin-share for fractionally subadditive valuations.

For deterministic allocations, to achieve a better approximation with the same technique, one idea would be to allocate more items to agents in the first phase. We believe it is promising that this extension gives a better approximation factor. However, extending the analysis is not trivial, and new ideas will be necessary to prove its correctness. A more fundamental obstacle lies in the fact that, regardless of the number of items allocated in the first phase, the approximation factor converges to 1/4. Therefore, developing new techniques is imperative to improve the approximation factor beyond 1/4.

For randomized allocations, our goal was to guarantee a fraction of MMS ex-ante, and a smaller fraction of MMS ex-post. Several interesting questions remain open. The most straight-forward direction is to improve approximation guarantees for both ex-ante and ex-post cases. None of our results are proved to be tight. Another notable point is that the running time of none of our algorithms is polynomially bounded. Are there efficient algorithms which achieve the same guarantees?

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
