# Randomized and Deterministic Maximin-share Approximations for Fractionally Subadditive Valuations

Hannaneh Akrami[1,3], Kurt Mehlhorn[1,4], Masoud Seddighin[2], and Golnoosh Shahkarami[1,3]

[1] *Max Planck Institute for Informatics*
[2] *Tehran Institute for Advanced Studies*
[3] *Graduiertenschule Informatik, Universität des Saarlandes*
[4] *Universität des Saarlandes*

## Abstract

We consider the problem of guaranteeing maximin-share (MMS) when allocating a set of indivisible items to a set of agents with fractionally subadditive (XOS) valuations. For XOS valuations, it has been previously shown that for some instances no allocation can guarantee a fraction better than $1/2$ of maximin-share to all the agents. Also, a deterministic allocation exists that guarantees 0.219225 of the maximin-share of each agent. Our results involve both deterministic and randomized allocations. On the deterministic side, we improve the best approximation guarantee for fractionally subadditive valuations to $3/13 = 0.230769$. We develop new ideas on allocating large items in our allocation algorithm which might be of independent interest. Furthermore, we investigate randomized algorithms and the Best-of-both-worlds fairness guarantees. We propose a randomized allocation that is 1/4-MMS ex-ante and 1/8-MMS ex-post for XOS valuations. Moreover, we prove an upper bound of $3/4$ on the ex-ante guarantee for this class of valuations.

## 1 Introduction

Fair allocation is a central problem in economics since decades. It arises naturally in real-world applications such as advertising, negotiation, rent sharing, inheritance, etc [14, 15, 16, 21, 29, 35]. In discrete fair division, the basic scenario is that we want to distribute a set $\mathcal{M}$ of $m$ indivisible items among $n$ agents, such that the allocation is deemed fair by the agents. Each agent $i$ has a valuation function $v_i : 2^{\mathcal{M}} \to \mathbb{R}^+$ that represents her happiness for receiving a subset of items.

How do we evaluate fairness? This question has been the subject of intense debates in various contexts, including philosophy, economics, distributive justice, and mathematics. Since the introduction of the cake-cutting[1] problem by Hugo Steinhaus in 1942 [33], scientists have suggested several different notions to evaluate fairness. The challenge is that a fairness notion must be both reasonable in terms of justice and implementable in practice. Conceived by Steinhaus, proportionality is one of the most natural and prominent notions. An allocation is *proportional* if the share allocated to every agent is worth at least $1/n$ of her value for the entire resource.

Unfortunately, despite many positive results on proportionality for cake-cutting, simple examples indicate that proportionality is not a proper fairness criterion for the case of indivisible items. For example, when there is one item and two agents, one agent receives nothing, though her proportional share is non-zero assuming that the item has positive value for both of the agents. An alternative form of proportionality adopted to deal with indivisibilities is *Maximin-share* (MMS)

---

[1]Cake-cutting is the continuous version of the fair allocation problem where the resource is a single divisible cake.

introduced by Budish [13]. For every agent $i$, the maximin-share of agent $i$, denoted by $\mathsf{MMS}_i$ is defined as follows:

$$\mathsf{MMS}_i = \max_{\langle \Pi_1, \Pi_2, \ldots, \Pi_n \rangle \in \Omega} \min_{1 \leq j \leq n} v_i(\Pi_j),$$

where $\Omega$ is the set of all partitions of $\mathcal{M}$ into $n$ parts. It is known that even when all the valuations are additive, $\mathsf{MMS}$ allocations, in which each agent $i$ gets at least $\mathsf{MMS}_i$, need not exist [19, 27, 31]. Nevertheless, several studies in recent years show that it is possible to guarantee a constant factor of her maximin-share to each agent for various classes of valuation functions, including additive, submodular, and fractionally subadditive.[2] See Table 1 for the state-of-the-art guarantees on the maximin-share.

| Valuation Class | Approximation Guarantee | Upper bound |
|---|---|---|
| Additive | $\frac{3}{4} + \frac{3}{3836}$ [1] | $\frac{39}{40}$ [19] |
| Submodular | $\frac{10}{27}$ [34] | $\frac{3}{4}$ [24] |
| Fractionally Subadditive | 0.219225 [32] | $\frac{1}{2}$ [24] |
| Subadditive | $\frac{1}{\log n \log \log n}$ [32] | $\frac{1}{2}$ [24] |

Table 1: A summary of the results for $\mathsf{MMS}$ in different valuation classes.

Let us revisit the instance with one item and two agents. Suppose that the item has value 6 for both agents. By definition, the proportional share of each agent is $6/2 = 3$, and since one agent receives no item, satisfying proportionality or any approximation of it is impossible. On the other hand, we have $\mathsf{MMS}_1 = \mathsf{MMS}_2 = 0$. Thus, allocating the item to any agent satisfies maximin-share. Indeed, we can circumvent the non-guaranteed existence of a fair allocation by reducing our expectation of fairness to the maximin-share. However, regardless of how we allocate the item, one agent receives one item, and the other receives nothing. Therefore, having one agent with zero utility is inevitable for any deterministic allocation in this example. The question then arises: Can we do better?

One way to improve the allocation is to use randomization and obtain a better guarantee in expectation (*ex-ante*). For example, we can allocate the item to each agent with probability $1/2$. This way, the expected utility of each agent is equal to 3. In economic terms, this allocation satisfies proportionality *ex-ante*. Note that one agent receives no item *ex-post* (that is, after fixing the outcome); however, it guarantees proportionality ex-ante to both agents.

Considering random allocations and ex-ante fairness makes the problem much handier. For instance, assuming there are $n$ items and $n$ agents, allocating each item to each agent with probability $1/n$ satisfies proportionality ex-ante. However, this randomized allocation has no ex-post fairness guarantee: with a non-zero probability, the outcome allocates all the items to one agent, and the rest of the agents receive no item. It is desirable to find allocations with simultaneous ex-ante and ex-post guarantees. The support of such an allocation is limited to outcomes with some desirable fairness guarantee. For example, consider the following random allocation: we choose a random permutation of the items and allocate the $i^{\text{th}}$ item in the permutation to agent $i$. This allocation satisfies proportionality ex-ante and maximin-share ex-post.

This approach is first introduced by Aziz [6]. Recently, several studies have investigated randomized allocations with both ex-ante and ex-post guarantees. Some notable results with the focus on additive valuations are (i) an ex-ante envy-free and ex-post EF1 allocation algorithm [22], (ii) an ex-ante proportional and ex-post 1/2-$\mathsf{MMS}$ allocation algorithm [9], and (iii) an ex-post 3/4-$\mathsf{MMS}$ and ex-ante 0.785-$\mathsf{MMS}$ allocation algorithm [3]. Very recently, Feldman *et al.* [20] studied best-

---

[2]We refer to Section 2 for a formal definition of these valuation classes.

of-both-worlds for subadditive valuations and gave an allocation algorithm with ex-ante guarantee of 1/2-envy-freeness and ex-post guarantee of 1/2-EFX and EF1.

In this paper, we explore fair deterministic and randomized allocations for fractionally subadditive valuation functions. A valuation function $v_i(\cdot)$ is fractionally subadditive (XOS), if there exists a family of additive valuation functions $u_{i,1}, u_{i,2}, \ldots, u_{i,\ell} : 2^{\mathcal{M}} \to \mathbb{R}_{\geq 0}$ such that for every set $S$ we have

$$v_i(S) = \max_{1 \leq k \leq \ell} u_{i,k}(S).$$

Fractionally subadditive is more general than additive, gross substitute, and submodular; it is less general than subadditive set functions.

Our fairness notion is maximin-share. We are looking for allocations that satisfy an approximation of maximin-share both ex-ante and ex-post for fractionally subadditive valuations. Of course, the ex-ante guarantee will be at least as strong as the ex-post one. We will also improve the ex-post guarantee (deterministically) for this class of valuations. We refer to the next subsection for an overview on our results and techniques.

## 1.1 Our Results and Techniques

In this paper, we provide improved approximation guarantees for the maximin-share in the fractionally subadditive setting. We investigate randomized and deterministic allocation algorithms. MMS is scale-invariant and in the rest of this paper, without loss of generality, we assume $\mathsf{MMS}_i = 1$ for all agents $i$.

### Randomized Allocations

For randomized allocations, we take one step toward extending the best-of-both-worlds idea for fairness concepts to valuations more general than additive. For the additive setting, Babaioff *et al.* [9] proved the existence of randomized allocations that are proportional ex-ante and 1/2-MMS ex-post. However, as we show in Section 3, though guaranteeing proportionality ex-ante is easy for valuations such as submodular, XOS, and subadditive, this notion is not always a proper choice as a fairness criterion. In fact, for some instances, proportionality can be as small as $O(1/n)$ of the MMS value, which is highly undesirable. Therefore, here we focus on guaranteeing MMS approximations both ex-ante and ex-post. More precisely, we are looking for randomized allocations that are $\alpha$-MMS ex-ante and $\beta$-MMS ex-post, where $0 < \beta < \alpha$. Very recently, Akrami et al. [3] studied the same question when all agents have additive valuations and proved the existence of a randomized allocation which is 3/4-MMS ex-post and 0.785-MMS ex-ante.

In contrast to the additive setting, for XOS valuations guaranteeing MMS ex-ante is not easy. Recall that in the additive setting, a fractional allocation that allocates a fraction $1/n$ of each item to each agent is proportional and consequently MMS. However, for some XOS instances this allocation is $O(1/n)$-MMS (Observation 3.2). Indeed, we show that there are instances for which guaranteeing MMS ex-ante is not possible. More precisely, we show that there are instances such that no randomized allocation can guarantee a factor better than 3/4 of her maximin-share to each agent (Lemma 3.3).

On the positive side, we propose an algorithm that finds a randomized allocation which is 1/4-MMS ex-ante. Furthermore, by leveraging additional innovative ideas, we extend this result to encompass both ex-ante and ex-post guarantees.

**Theorem 1.1.** *For any instance with* XOS *valuations, there exists a randomized allocation that is* 1/4-MMS *ex-ante.*

The idea to prove the approximation guarantee of our allocation is inspired by the work of Ghodsi *et al.* [24]. In fact, we show that the fractional allocation $\mathcal{F}$ which is obtained from the following program is 1/4-MMS:

$$
\begin{aligned}
\text{maximize} \quad & \sum_{1 \leq i \leq n} u_i \\
\text{subject to} \quad & \sum_{1 \leq i \leq n} f_{i,j} = 1 & \forall_j \\
& f_{i,j} \geq 0 & \forall_{i,j} \\
& u_i = \min\left(\frac{1}{2}, \max_k \sum_j u_{i,k}(\{b_j\}) f_{i,j}\right). & \forall_i
\end{aligned}
\tag{1}
$$

Intuitively, Program 1 defines an alternative valuation function $\bar{v}_i(\cdot)$ for each agent $i$ and then finds an allocation that maximizes social welfare with respect to these valuations. For every $i$, $\bar{v}_i(\cdot)$ is the same as $v_i(\cdot)$, except that for every bundle $X$ with $v_i(X) > 1/2$ we have $\bar{v}_i(X) = 1/2$. From an economical standpoint, one can see the answer of this program as an interesting trade-off between fairness and social welfare.

We can then convert the fractional allocation $\mathcal{F}$ into a randomized one (Theorem 2.9 and Lemma 2.5). However, there is no non-trivial guarantee on the fairness of the ex-post allocation. To resolve this issue, we add one additional step to our algorithm (namely allocating single large items), and also add another constraint to the optimization program. Before solving the optimization problem, we check to see if a single item can satisfy an agent. The main goal of this step is to make sure that the value of each remaining item for the remaining agents is small enough so that we can use Theorem 2.9 to convert the fractional allocation into a randomized one with an ex-post guarantee.

We also add another constraint to the optimization problem to obtain the following half-integral optimization program:

$$
\begin{aligned}
\text{maximize} \quad & \sum_{1 \leq i \leq n} u_i \\
\text{subject to} \quad & \sum_{1 \leq i \leq n} f_{i,j} = 1 & \forall_j \\
& f_{i,j} \in \{0, 1/2, 1\} & \forall_{i,j} \\
& u_i = \min\left(\frac{1}{2}, \max_k \sum_j u_{i,k}(\{b_j\}) f_{i,j}\right). & \forall_i
\end{aligned}
\tag{2}
$$

Despite this additional step and constraint, we prove that the answer of Program 2 also gives the 1/4-MMS ex-ante approximation guarantee. However, note that the upper bound on the value of items obtained from the additional step combined with Theorem 2.9 still gives no approximation guarantee better than 0 for the ex-post allocation, because there might be some items with value close to 1/4 in the bundle of agents. To prove the ex-post guarantee, we provide a more intricate analysis of the method that is used in Theorem 1.1 and show that our allocation is 1/8-MMS ex-post. The fact that the allocation is half-integral plays a key role in the proof. The pseudocode of our approach is shown in Algorithm 1.

**Theorem 1.2.** *For any instance with* XOS *valuations, Algorithm 1 returns a randomized allocation that is* 1/4-MMS *ex-ante and* 1/8-MMS *ex-post.*

### Deterministic Allocations

Since the impossibility result on MMS by Proccacia and Wang [31], there has been considerable work of establishing approximate MMS guarantees for various classes of valuation functions [1, 2, 11, 23, 24, 27, 32]. The best previous deterministic guarantee for XOS valuations was 0.219225 [32]. We improve the guarantee to $3/13 \approx 0.2307$. In order to do so, like in our randomized algorithm, we first allocate large items. However, in addition to allocating single large items, here we also allocate pairs and triples of items if they satisfy some agent up to $3/13$ factor of their MMS value. This way we get a stronger upper bound on the value of most of the remaining items.

In the last step, we output an allocation which maximizes the social welfare with respect to valuations $\bar{v}_i(\cdot) = \min\{\frac{6}{13}, v_i(\cdot)\}$ for the remaining agents and items. Note that this last step is also in correspondence with the last step of the randomized algorithm. Here we cap the value of the bundles with $6/13$ (instead of $1/2$) and we output an integral allocation with maximum social welfare (instead of a half-integral allocation with maximum social welfare). Moreover, here we need to do a more careful analysis to show that the output is indeed a $3/13$-MMS allocation. The pseudocode of our approach is shown in Algorithm 2.

**Theorem 1.3.** *For any instance with* XOS *valuations, Algorithm 2 returns a* $3/13$-MMS *allocation.*

## 1.2 Further Related Work

**Maximin-share fairness notion.** Budish [13] introduced the MMS-concept. As already mentioned, there are instances with additive valuations for which no MMS allocation exists [19, 27, 31]. Therefore, several studies considered the approximation guarantees. For additive valuations, the first approximation guarantee, 2/3-MMS, is given by Kurokawa et al. [27]. The approximation factor improved over time by Ghodsi et al. [24] to $3/4$, and then to $3/4 + o(1)$ by Garg and Taki [23] and Akrami et al. [2]. The best-known result for this valuation class is $3/4 + 3/3836$ which was very recently obtained by Akrami and Garg [1]. Moreover, there are several works on MMS approximations for submodular valuations [11, 24], and subadditive valuations [24, 32]. Also, Farhadi et al. [17] proved the tight bound of $1/n$-MMS for the setting where agents can have different entitlements.

**Fractionally subadditive valuations.** For fractionally subadditive valuations, Ghodsi et al. [24] gave a 1/5-MMS approximation algorithm; Seddighin et al. [32] improved the factor to $1/4.6 = 0.2173913$. A special case of fractionally subadditive valuations has been studied by Li and Vetta [28]. Also other notions of fairness have been studied for this class of valuation. Feige [18] studied maximizing social welfare. Also, Hoefer et al. [26] proved the existence of randomized allocations which are proportional ex-ante and proportional up to one item (PROP1) ex-post. Furthermore, the notion of envy-freeness up to any item (EFX) has been considered by Plaut and Roughgarden [30].

**Best-of-both-worlds fairness.** Aleksandrov et al. [4] first explored this line of research in the context of the food bank problem for a special case of additive valuations. Freeman et al. [22] proposed a randomized polynomial-time algorithm for additive valuations that is envy-free ex-ante and EF1 ex-post. Afterward, Aziz [6] modified this algorithm to also get the Probabilistic Serial fractional outcome [12] as well as a weak notion of efficiency with a simpler proof. Babaioff et al. [9] provided an allocation algorithm in which the expected value of each agent's bundle is at least a $1/n$-fraction of her value for the set of all items (ex-ante proportional) and ex-post proportional up to one item and 1/2-MMS allocations for the additive valuations. Feldman et al. [20] gave an allocation algorithm for subadditive valuations with guarantees of 1/2-envy-freeness ex-ante and

1/2-EFX and EF1 ex-post. Moreover, studies have been conducted for additive valuations with binary marginals [6, 25], and for matroid rank valuations [7, 8].

## 2  Preliminaries

A fair division instance is denoted by $\mathcal{I} = (\mathcal{N}, \mathcal{M}, \mathcal{V})$ where $\mathcal{N} = [n]$ is a set of $n$ agents, $\mathcal{M}$ is a set of $m$ indivisible items and $\mathcal{V} = (v_1, \ldots, v_n)$ is a vector of valuation functions. Each agent $i$ has a valuation function $v_i : 2^{\mathcal{M}} \to \mathbb{R}_{\geq 0}$ that represents her value for every bundle of items. Thus, for every set $S$ of items, $v_i(S)$ shows the value of $S$ to agent $i$. We denote the $j^{\text{th}}$ item by $b_j$ and write $v_i(b_j)$ for the value $v_i(\{b_j\})$ of $b_j$ to agent $i$. An allocation of items is a partition of $\mathcal{M}$ into $n$ parts (i.e., bundles) where the $i^{\text{th}}$ bundle is the share allocated to agent $i$. For an allocation $\mathcal{A}$, we denote the bundle of agent $i$ by $\mathcal{A}_i$.

We assume that the valuations are monotone, i.e., for every two sets $S$ and $T$ such that $S \subseteq T$, we have $v_i(S) \leq v_i(T)$, and normalized, i.e., $v_i(\emptyset) = 0$ for all $i \in \mathcal{N}$. Our discussion in this paper involves two classes of valuation functions: additive and fractionally subadditive valuation functions. A valuation function $v_i(\cdot)$ is additive, if for every set $S$ of items, we have

$$v_i(S) = \sum_{b_j \in S} v_i(b_j).$$

**Definition 2.1** (XOS)**.** A valuation function $v_i(\cdot)$ is fractionally subadditive (XOS), if there exists a family of additive valuation functions $u_{i,1}, u_{i,2}, \ldots, u_{i,\ell} : 2^{\mathcal{M}} \to \mathbb{R}_{\geq 0}$ such that for every set $S$ we have

$$v_i(S) = \max_{1 \leq k \leq \ell} u_{i,k}(S).$$

Given an allocation $\mathcal{A}$, we denote by $u_{i,i'}$, an additive function of $v_i$ that defines $v_i(\mathcal{A}_i)$, i.e., $v_i(\mathcal{A}_i) = u_{i,i'}(\mathcal{A}_i)$. Another term we frequently use in this paper is contribution; which is defined as the marginal value of one set to another.

**Definition 2.2** (Contribution)**.** For every sets $S, T$ of items such that $S \subseteq T$, we define the marginal contribution of $S$ to $T$ with respect to valuation function $v$, denoted by $C_v^T(S)$, as follows:

$$C_v^T(S) = v(T) - v(T \setminus S)$$

i.e., the marginal contribution of $S$ to $T$ is the value decrease when $S$ is removed from $T$.

**Example 2.3.** Consider 5 items $b_1, b_2, \ldots, b_5$, and an identical valuation $v$ for all the agents. Suppose that $v$ is a fractionally subadditive function consisting of two additive function $u_1 = [2, 8, 4, 5, 1]$ and $u_2 = [5, 1, 9, 4, 5]$ (the $j^{\text{th}}$ element is the value for $b_j$). For set $S = \{b_1, b_2, \ldots, b_5\}$ we have $u_1(S) = 20$ and $u_2(S) = 24$. Hence, $v(S) = \max(u_1(S), u_2(S)) = 24$. Also, the marginal contribution of item $b_3$ to set $S$ is $C_v^S(\{b_3\}) = v(S) - v(S \setminus \{b_3\}) = 24 - 16 = 8$, which is smaller than $u_2(b_3) = 9$.

With abuse of notation, for an allocation $\mathcal{A}$ of items to agents with valuation vector $\mathcal{V} = (v_1, \ldots, v_n)$ and every set $S$ of items, we define the contribution of $S$ to $\mathcal{A}$ with respect to $\mathcal{V}$, denoted by $C_{\mathcal{V}}^{\mathcal{A}}(S)$ as follows:

$$C_{\mathcal{V}}^{\mathcal{A}}(S) = \sum_{1 \leq i \leq n} C_{v_i}^{\mathcal{A}_i}(\mathcal{A}_i \cap S). \tag{3}$$

However, since an XOS valuation function might include many additive functions, Equality (3) is not always practical. Therefore, we use Observation 2.4 to bound $C_{\mathcal{V}}^{\mathcal{A}}(S)$. Since $u_{i,i'}(\mathcal{A}_i) = u_{i,i'}(\mathcal{A}_i \setminus S) + u_{i,i'}(\mathcal{A}_i \cap S)$ and we have $v_i(\mathcal{A}_i) - v_i(\mathcal{A}_i \setminus S) \leq u_{i,i'}(\mathcal{A}_i) - u_{i,i'}(\mathcal{A}_i \setminus S) = u_{i,i'}(\mathcal{A}_i \cap S)$. Summing over $i$ yields the following Observation.

**Observation 2.4.** For every allocation $\mathcal{A}$ of items to agents with valuation vector $\mathcal{V}$ and every set $S$ of items, we have

$$C_{\mathcal{V}}^{\mathcal{A}}(S) \leq \sum_{1 \leq i \leq n} u_{i,i'}(\mathcal{A}_i \cap S).$$

Our goal is to allocate the items to the agents in a fair manner. Here we discuss two share-based notions of fairness, namely proportionality, and maximin-share. For agent $i$, we define the proportional share of agent $i$, denoted by $\pi_i$ as:

$$\pi_i = v_i(\mathcal{M})/n.$$

We also define the maximin-share of agent $i$, denoted by $\mathsf{MMS}_i$ as

$$\mathsf{MMS}_i = \max_{\langle \Pi_1, \Pi_2, \dots, \Pi_n \rangle \in \Omega} \min_{1 \leq j \leq n} v_i(\Pi_j),$$

where $\Omega$ is the set of all partitions of $\mathcal{M}$ into $n$ bundles. Moreover, if for a partition $\Pi = \langle \Pi_1, \Pi_2, \dots, \Pi_n \rangle$ of $\mathcal{M}$ into $n$ bundles we have $v_i(\Pi_j) \geq \mathsf{MMS}_i$ for all $j \in [n]$, we say $\Pi$ is an "MMS partition" or "optimal partition" of agent $i$. By definition of $\mathsf{MMS}$, for every agent $i$ there exists at least one partition $\Pi$ which is an $\mathsf{MMS}$ partition for agent $i$. For brevity, in the rest of the paper, we assume that the valuations are scaled so that for every agent $i$, we have $\mathsf{MMS}_i = 1$. We also define approximate versions of these two notions as follows. For any $\alpha > 0$, we say an allocation is $\alpha$-proportional, if it guarantees to each agent $i$ a bundle with value at least $\alpha \pi_i$. Likewise, in an $\alpha$-$\mathsf{MMS}$ allocation, the value of the share allocated to each agent $i$ is at least $\alpha$.

**Randomized allocation.** In this paper, we also consider randomized allocations. A randomized allocation is a distribution over a set of deterministic allocations. For a randomized allocation $\mathcal{R}$, we denote by $D(\mathcal{R})$ the set of allocations in the support of $\mathcal{R}$. For a randomized allocation $\mathcal{R}$, the expected welfare of agent $i$ is defined as

$$v_i(\mathcal{R}) = \sum_{\mathcal{A} \in D(\mathcal{R})} v_i(\mathcal{A}_i) \cdot p_{\mathcal{A}},$$

where $p_{\mathcal{A}}$ is the probability of allocation $\mathcal{A}$ in $\mathcal{R}$.

**Fractional allocation.** En route to proving our results, we leverage another relaxed form of allocation called fractional allocation. In a fractional allocation, we ignore the indivisibility assumption and treat each item as a divisible one. Formally, a fractional allocation $\mathcal{F}$ is a set of $nm$ variables $f_{i,j}$ indicating the fraction of item $b_j$ allocated to agent $i$. Therefore, we expect allocation $\mathcal{F}$ to satisfy the following constraints:

$$\forall_j \qquad \sum_{1 \leq i \leq n} f_{i,j} \leq 1 \qquad \qquad \text{(we have one unit of each item)}$$

$$\forall_{i,j} \qquad 0 \leq f_{i,j} \qquad \text{(each agent receives a non-negative share of each item)}$$

A fractional allocation is *complete* if $\sum_i f_{i,j} = 1$ for all items $b_j$, i.e., all items are completely allocated. Given a fractional allocation $\mathcal{F}$, we define the utility of agent $i$ for $\mathcal{F}$ in the same way as we calculate it for integral allocations:

$$v_i(\mathcal{F}) = \max_k \sum_{1 \leq j \leq m} u_{i,k}(b_j) f_{i,j}.$$

Complete fractional allocations give rise to randomized allocations in the standard way, i.e., the probability $p_{\mathcal{A}}$ of an allocation $\mathcal{A}$ is defined as

$$p_{\mathcal{A}} = \prod_{1 \leq i \leq n} \prod_{b_j \in \mathcal{A}_i} f_{ij}. \tag{4}$$

Then $\sum_{\mathcal{A}} p_{\mathcal{A}} = 1$. Indeed, view an integral allocation $\mathcal{A}$ as a mapping $\pi$ from $[m]$ to $[n]$: $\pi(j) = i$ iff $b_j \in \mathcal{A}_i$. Then $\sum_{\mathcal{A}} p_{\mathcal{A}} = \sum_{\mathcal{A}} \prod_i \prod_{b_j \in \mathcal{A}_i} f_{ij} = \sum_{\pi \in [n]^{[m]}} \prod_j f_{\pi(j)j} = \prod_j (\sum_i f_{ij}) = \prod_j 1 = 1$.

**Lemma 2.5.** *Let $\mathcal{F}$ be a complete fractional allocation and let randomized allocation $\mathcal{R}$ be defined by (4). Then for* XOS *valuation functions $v_i$,*

$$v_i(\mathcal{R}) \geq v_i(\mathcal{F})$$

*for all $1 \leq i \leq n$.*

**Proof.** For each $i$, let $i'$ be such that $u_{i,i'}(\mathcal{F}) = \max_k u_{i,k}(\mathcal{F})$. We view again an allocation $\mathcal{A}$ as a mapping $\pi \in [n]^{[m]}$. Then

$$
\begin{aligned}
v_i(\mathcal{R}) &= \sum_{\mathcal{A}} v_i(\mathcal{A}_i) p_{\mathcal{A}} \\
&= \sum_{\pi} v_i(\pi^{-1}(i)) \prod_t f_{\pi(t),t} \\
&\geq \sum_{\pi} \sum_{j;\ \pi(j)=i} u_{i,i'}(j) \prod_t f_{\pi(t),t} \\
&= \sum_j \sum_{\pi;\ \pi(j)=i} u_{i,i'}(j) f_{i,j} \prod_{t \neq j} f_{\pi(t),t} \\
&= \sum_j u_{i,i'}(j) f_{i,j} \sum_{\pi \in [n]^{[m]\setminus j}} \prod_{t \neq j} f_{\pi(t),t} \\
&= \sum_j u_{i,i'}(j) f_{i,j} \prod_{t \neq j} \sum_{\ell} f_{\ell,t} \\
&= \sum_j u_{i,i'}(j) f_{i,j} \prod_{t \neq j} 1 \\
&= \sum_j u_{i,i'}(j) f_{i,j} \\
&= v_i(\mathcal{F}).
\end{aligned}
$$

$\square$

We also need to redefine *contribution* for fractional allocations and fractional bundles. Suppose that $\mathcal{F}$ is a fractional allocation and $S$ is a fractional set of items. Since items are fractionally allocated, the term contribution must be defined more precisely. For example, suppose that set $S$ consists of a fraction 0.4 of item $b_j$, and in allocation $\mathcal{F}$, 0.2 of $b_j$ belongs to agent $i_1$, 0.5 of $b_j$ belongs to agent $i_2$, and 0.3 of $b_j$ belongs to agent $i_3$. We need to define exactly how the 0.4 fraction of item $b_j$ in $S$ is distributed over the agents. One reasonable strategy is to choose the share of each agent in a way that after removal of $S$ from $\mathcal{F}$ we have the smallest possible decrease in the social welfare. Based on this strategy, assuming that $s_j$ is the fraction of item $b_j$ in $S$, we

define the contribution of $S$ to $\mathcal{F}$, denoted by $C_{\mathcal{V}}^{\mathcal{F}}(S)$ as the answer of the following optimization program:

$$\text{minimize} \quad \sum_{1 \leq i \leq n} v_i(\mathcal{F}) - v_i(\mathcal{F}')$$

$$\text{subject to} \quad \sum_{1 \leq i \leq n} f_{i,j} - f'_{i,j} = s_j \qquad\qquad \forall j$$

$$0 \leq f'_{i,j} \leq f_{i,j} \qquad\qquad \forall i,j \qquad (5)$$

Generally, it is hard to deal with the above optimization program. Here, we use an important property of $C_{\mathcal{V}}^{\mathcal{F}}(\cdot)$ to obtain our results.

**Lemma 2.6.** *Let $\mathcal{F}$ be an arbitrary fractional allocation and assume that for every agent $i$, $v_i(\cdot)$ is* XOS. *Then, for every partition of the items into fractional sets $S_1, S_2, \ldots, S_t$, we have*

$$\sum_{1 \leq k \leq t} C_{\mathcal{V}}^{\mathcal{F}}(S_k) \leq \sum_{1 \leq i \leq n} v_i(\mathcal{F}).$$

**Proof.** For every agent $i$, let $i' = \arg\max_k u_{i,k}(\mathcal{F})$. By definition, we have

$$v_i(\mathcal{F}) = \sum_{1 \leq j \leq m} u_{i,i'}(b_j) \cdot f_{i,j}.$$

We will define for each set $S_k$ an allocation $\mathcal{F}^{(k)}$ by reducing the allocation $\mathcal{F}$ proportionally, i.e., we will replace $f_{ij}$ by $(1 - s_{k,j})f_{ij}$, where $s_{k,j}$ is the fraction of item $j$ belonging to set $S_k$. Note that since $S_1, S_2, \ldots, S_t$ is a partition of items, $\sum_k s_{k,j} = 1$. For every $1 \leq k \leq t$, $1 \leq i \leq n$, and $1 \leq j \leq m$, define variable $f'^{(k)}_{i,j}$ as follows:

$$f'^{(k)}_{i,j} = (1 - s_{k,j}) \cdot f_{i,j}.$$

Denote by $\mathcal{F}'^{(k)}$ the partial allocation defined by the variables $f'^{(k)}_{i,j}$. Since for every $j$, we have

$$\sum_{1 \leq i \leq n} (f_{i,j} - f'^{(k)}_{i,j}) = \sum_{1 \leq i \leq n} s_{k,j} f_{i,j}$$

$$= s_{k,j},$$

$\mathcal{F}'^{(k)}$ is a feasible solution to Program (5). Therefore, we have

$$\sum_{1 \leq k \leq t} C_{\mathcal{V}}^{\mathcal{F}}(S_k) \leq \sum_{1 \leq k \leq t} \sum_{1 \leq i \leq n} (v_i(\mathcal{F}) - v_i(\mathcal{F}'^{(k)}))$$

$$\leq \sum_{1 \leq k \leq t} \sum_{1 \leq i \leq n} \sum_{1 \leq j \leq m} u_{i,i'}(b_j)(f_{i,j} - f'^{(k)}_{i,j})$$

$$= \sum_{1 \leq i \leq n} \sum_{1 \leq j \leq m} \sum_{1 \leq k \leq t} u_{i,i'}(b_j) s_{k,j} f_{i,j}$$

$$= \sum_{1 \leq i \leq n} \sum_{1 \leq j \leq m} u_{i,i'}(b_j) f_{i,j}$$

$$= \sum_{1 \leq i \leq n} v_i(\mathcal{F}).$$

$\square$

## 2.1 Ex-ante and Ex-post Fairness Guarantees.

For a randomized allocation, we define two types of fairness guarantees, namely *ex-ante* and *ex-post* as in Definitions 2.7 and 2.8.

**Definition 2.7** (ex-ante). Given a randomized allocation $\mathcal{R}$, we say $\mathcal{R}$ is $\alpha$-MMS *ex-ante*, if for every agent $i$, we have $v_i(\mathcal{R}) \geq \alpha$. Similarly, $\mathcal{R}$ is $\alpha$-proportional, if for every agent $i$, $v_i(\mathcal{R}) \geq \alpha\pi_i$.

**Definition 2.8** (ex-post). An allocation $\mathcal{R}$ is $\alpha$-MMS *ex-post*, if every allocation $\mathcal{A} \in D(\mathcal{R})$ is $\alpha$-MMS. Similarly, we say $\mathcal{R}$ is $\alpha$-proportional ex-post if every allocation $\mathcal{A} \in D(\mathcal{R})$ is $\alpha$-proportional.

One tool that we refer to in this paper is the result of Babaioff, Ezra, and Feige for converting a fractional allocation into a faithful randomized allocation [9].

**Theorem 2.9** (Proved in [9]). *Assume that the valuations are additive and let $\mathcal{F}$ be a fractional allocation. Then there exists a randomized allocation $\mathcal{R}$ such that the ex-ante utility of the agents for $\mathcal{R}$ is the same as the utility of the agents in $\mathcal{F}$, and for every allocation $\mathcal{A}$ in the support of $\mathcal{R}$ the following holds:*

$$\forall_i, v_i(\mathcal{A}_i) \geq v_i(\mathcal{R}) - \max_{j: f_{i,j} \notin \{0,1\}} v_i(b_j).$$

In this paper, we use a more delicate analysis of the method used in Theorem 2.9 to convert fractional allocations into randomized ones. This helps us improve our ex-post approximation guarantee.

## 3 Ex-ante Guarantees

In this section, our goal is to explore the possibility of designing randomized allocations that is $\alpha$-MMS ex-ante or $\alpha$-proportional ex-ante. Note that, in contrast to the additive case, for XOS valuations there is no meaningful correspondence between proportionality and maximin-share; proportional share can be larger or smaller than maximin-share. Recall that for the additive case, we always have $\pi_i \geq \mathsf{MMS}_i$ and therefore, maximin-share is implied by proportionality. However, for fractionally subadditive valuations, $\pi_i$ can be as small as $\mathsf{MMS}_i/n$.

For the additive setting, a simple fractional allocation that allocates a fraction $1/n$ of each item to each agent guarantees proportionality and consequently maximin-share. Using Theorem 2.9 one can convert this allocation to a randomized allocation that is proportional ex-ante. In Observation 3.1 we show that proportionality can be guaranteed ex-ante for XOS valuations.[3]

**Observation 3.1.** Every randomized allocation that allocates each item with probability $1/n$ to each agent is proportional ex-ante.

**Proof.** Let $\mathcal{R}$ be a randomized allocation such that the probability that item $b_j$ is allocated to agent $i$ is $1/n$. Also, let $u_{i,i'}$ be the additive valuation function that defines $v_i(\mathcal{M})$, i.e., $v_i(\mathcal{M}) =$

---

[3]Note that for now, we are not concerned about the ex-post guarantee of our allocation.

$u_{i,i'}(\mathcal{M})$, and let $p_{\mathcal{A}}$ be the probability that allocation $\mathcal{A}$ is chosen in $\mathcal{R}$. We have

$$
\begin{aligned}
v_i(\mathcal{R}) &= \sum_{\mathcal{A}\in D(\mathcal{R})} p_{\mathcal{A}} v_i(\mathcal{A}_i) \\
&\geq \sum_{\mathcal{A}\in D(\mathcal{R})} p_{\mathcal{A}} u_{i,i'}(\mathcal{A}_i) \\
&= \sum_{\mathcal{A}\in D(\mathcal{R})} p_{\mathcal{A}} \sum_{b_j \in \mathcal{A}_i} u_{i,i'}(b_j) \\
&= \sum_{1\leq j\leq m} \sum_{\mathcal{A}_i \ni b_j} u_{i,i'}(b_j) p_{\mathcal{A}} \\
&= \sum_{1\leq j\leq m} u_{i,i'}(b_j)/n \\
&= v_i(\mathcal{M})/n.
\end{aligned}
$$

$\square$

In contrast to the additive setting, finding an allocation that guarantees maximin-share is not trivial. Indeed, the simple fractional allocation that guarantees proportionality in Observation 3.1 can be as bad as $O(1/n)$-MMS.

**Observation 3.2.** Let $\mathcal{F}$ be a fractional allocation that allocates a fraction $1/n$ of each item to each agent. Then, there exists an instance such that the maximin-share guarantee of $\mathcal{F}$ is $O(1/n)$.

**Proof.** Consider the following instance: there are $n^2$ items. The valuation of agent $i$ is an XOS set function consisting of $n$ additive valuation functions as follows: partition the items into $n$ bundles each with $n$ items. For each additive function $u_{i,k}$, the value of each item in the $k^{\text{th}}$ bundle is $1/n$ and the value of the rest of the items is 0. It is easy to observe that for this instance, the MMS value of each agent is 1, and the value of each agent for her bundle in $\mathcal{F}$ is $1/n$. $\square$

Generally, there are two main challenges in the process of designing a randomized allocation that guarantees an approximation of maximin-share. In contrast to the additive setting, finding a fractional or randomized allocation that approximates maximin-share is not easy. As well as that, transforming a fractional allocation into a randomized one is not straightforward. Indeed, as we show in Lemma 3.3, neither fractional allocations nor randomized allocations can guarantee MMS. We prove an upper bound on the best approximation guarantee of each one of these allocation types.

**Lemma 3.3.** *For* XOS *valuations, the best* MMS *guarantee for fractional allocations and the best ex-ante* MMS *guarantee for randomized allocation is upper bounded by* 3/4.

**Proof.** Consider the following instance: there are two agents and four items. The fractionally subadditive valuation of each agent consists of 2 additive functions. The valuations are as follows:

$$
\begin{aligned}
u_{1,1}(\{b_1\}) &= u_{1,1}(\{b_2\}) = 1, u_{1,1}(\{b_3\}) = u_{1,1}(\{b_4\}) = 0, \\
u_{1,2}(\{b_1\}) &= u_{1,2}(\{b_2\}) = 0, u_{1,2}(\{b_3\}) = u_{1,2}(\{b_4\}) = 1, \\
u_{2,1}(\{b_1\}) &= u_{2,1}(\{b_4\}) = 1, u_{2,1}(\{b_2\}) = u_{2,1}(\{b_3\}) = 0, \\
u_{2,2}(\{b_1\}) &= u_{2,2}(\{b_4\}) = 0, u_{2,2}(\{b_2\}) = u_{2,2}(\{b_3\}) = 1.
\end{aligned}
$$

It is easy to check that for the above instance, the maximin-share of each agent is equal to 2, and no fractional allocation can guarantee more than 1.5 to both agents. Also, we can guarantee a

value of 1.5 to both agents by giving the first and the third items receptively to agents 1 and 2, and giving half of the remaining items to each agent.

Now, we show that the same upper bound also holds for the ex-ante guarantee of randomized allocations. Assume that $\mathcal{R}$ is the randomized allocation that maximizes the maximin-share guarantee for this instance. Since there are two agents, we know that $\mathcal{R}$ maximizes the following objective:

$$\alpha = \min \left( \sum_{S \subseteq \mathcal{M}} \mathbb{P}(S) v_1(S), \sum_{S \subseteq \mathcal{M}} \mathbb{P}(S) v_2(\mathcal{M} \setminus S) \right),$$

where $\mathbb{P}(S)$ is the probability that set $S$ is allocated to agent 1. Furthermore, since for every integers $y, z$ we have $\min(y, z) \leq (y + z)/2$, and

$$
\begin{aligned}
\alpha &\leq \left( \sum_{S \subseteq \mathcal{M}} \mathbb{P}(S) v_1(S) + \sum_{S \subseteq \mathcal{M}} \mathbb{P}(S) v_2(\mathcal{M} \setminus S) \right) /2 \\
&= \sum_{S \subseteq \mathcal{M}} \mathbb{P}(S) \left( v_1(S) + v_2(\mathcal{M} \setminus S) \right) /2.
\end{aligned}
$$

One can easily check that for every set $S$, the value of $v_1(S) + v_2(\mathcal{M} \setminus S)$ is upper bounded by 3. Therefore, we have

$$
\begin{aligned}
\alpha &\leq \sum_{S \subseteq \mathcal{M}} (3/2) \mathbb{P}(S) \\
&\leq 3/2.
\end{aligned}
$$

Hence, the best possible approximation guarantee for MMS in this instance is at most $1.5/2 = 3/4$. Note that one can guarantee a value of 1.5 ex-ante to both the agents by giving the first and the third items receptively to agents 1 and 2, and giving the rest of the items with a probability of $1/2$ to one of the agents. $\square$

Before we prove our lower bound on the maximin-share guarantee for randomized allocations, we note that another challenge about XOS valuations is that in sharp contrast to additive valuations, transforming a fractional allocation to a randomized one is not easy. Indeed, we can show that for a fractional allocation $\mathcal{F}$ there might be randomized allocations $\mathcal{R}$ and $\mathcal{R}'$ with different utility guarantees for the agents, such that in both $\mathcal{R}$ and $\mathcal{R}'$ the probability that each item $b_j$ is allocated to agent $i$ is equal to $f_{i,j}$. Example 3.4 gives more insight into this challenge.

**Example 3.4.** Consider the instance described in the proof of Observation 3.2 and define allocations $\mathcal{R}$ and $\mathcal{R}'$ as follows:

- Allocation $\mathcal{R}$ allocates each item to each agent with probability $1/n$.

- Allocation $\mathcal{R}'$ considers a random permutation of the bundles in the optimal MMS-partitioning of the agents and allocates the $i^{\text{th}}$ bundle in the permutation to agent $i$.[4]

It is easy to check that in both of these allocations, each item is allocated to each agent with probability $1/n$. However, the maximin-share guarantee of $\mathcal{R}$ is $O(\log n/n)$. To show this, one can argue that using Chernoff bound the probability that more than $3 \log n$ items from the same bundle

---

[4]Note that in the instance described in Observation 3.2 the optimal MMS-partitioning of all the agents are the same.

in the optimal partition are allocated to agent $i$ is $O(1/n^2)$. Hence, the expected value of agent $i$ for her share is at most

$$1 \cdot \frac{1}{n^2} + \frac{3 \log n}{n} \cdot \frac{n^2 - 1}{n^2} \leq \frac{4 \log n}{n}.$$

On the other hand, allocation $\mathcal{R}'$ guarantees value 1 to all the agents.

Despite these hurdles, in Theorem 1.1 we show that there exists a randomized allocation that guarantees 1/4-MMS to all the agents ex-ante. To prove Theorem 1.1, we first show that a fractional allocation exists that is 1/4-MMS. Next, we convert it to a randomized allocation. Theorem 1.1 along with Lemma 3.3 leave a gap of $[1/4, 3/4]$ between the best upper bound and the best lower-bound for the maximin-share guarantee of fractional allocations in the XOS setting.

**Theorem 1.1.** *For any instance with* XOS *valuations, there exists a randomized allocation that is* 1/4-MMS *ex-ante.*

**Proof Idea:** Let $\mathcal{F}$ be the a (fractional) allocation maximizing social welfare and assume that there is an agent $i$ whose bundle has value less than 1/4 to her. Now consider the MMS-partition of agent $i$. Each bundle in this partition has value at least 1 to $i$. We can split each bundle into two so that each subbundle has value at least 1/2 to $i$. Let $B$ be one of these $2n$ subbundles. Imagine that we reassign the items in $B$. We take away the items in $B$ from their current owners and give them to $i$. Then $i$ would gain more than 1/4, but the other agents would lose. The loss is bounded by $C_{\mathcal{F}}(B)$. Why should this quantity be less than 1/4 for one of the $2n$ subbundles?

Lemma 2.6 comes to the rescue. We have

$$\sum_{1 \leq j \leq 2n} C_{\mathcal{V}}^{\mathcal{F}}(B_j) \leq \sum_{1 \leq i \leq n} v_i(\mathcal{F}),$$

where $B_1$ to $B_{2n}$ are the subbundles. If the right hand side is strictly less than $n/2$, the desired subbundle exists. We can achieve this by replacing our valuations $v_i$ by valuations $\bar{v}_i$ that assign no set a value more than 1/2.

**Proof.** For a fractional allocation, we define the truncated value of agent $i$, denoted by $\bar{v}_i$, of a fractional set $S$ as follows:

$$\bar{v}_i(S) = \min\left(\frac{1}{2}, \max_k \sum_{1 \leq j \leq m} u_{i,k}(b_j) s_j\right). \tag{6}$$

where $u_{i,k}$ is the $k^{\text{th}}$ additive function of $v_i$ and $s_j$ is the fraction of item $b_j$ that belongs to set $S$. If the valuation $v_i(\cdot)$ is XOS, then $\bar{v}_i(\cdot)$ is also XOS [24]. Let $\bar{v} = (\bar{v}_1, \ldots, \bar{v}_n)$.

Now let $\mathcal{F}$ be the complete fractional allocation that maximizes

$$Z = \sum_{1 \leq i \leq n} \bar{v}_i(\mathcal{F}). \tag{7}$$

We know $Z \leq n/2$ since for any fractional bundle $S$ and every agent $i$, we have $\bar{v}_i(S) \leq 1/2$. We claim that $\mathcal{F}$ allocates each agent a bundle with a value of at least 1/4. For the sake of a contradiction, assume that this is not true and let agent $a$ be an agent whose share is worth less than 1/4 to her. Then $Z < n/2 - 1/4$.

Given that the maximin-share of agent $i$ is equal to 1, she can divide the items in $\mathcal{M}$ into $2n$ fractional bundles, each of which has a value of at least 1/2 to her.

**Claim 3.5.** *Since the maximin-share of agent $i$ is at least 1, she can divide the item in $\mathcal{M}$ into $2n$ fractional bundles, each with value at least 1/2 to her.*

*Proof.* Consider the optimal maximin-share partition of agent $i$, and for each bundle in this partition divide that bundle into two fractional sub-bundles with a value of at least $1/2$. Since the valuation of agent $i$ is XOS, such a division is always possible: just take a fractional sub-bundle with a value of exactly $1/2$ from each bundle. The remaining (fractional) items in that bundle also form a sub-bundle with a value of at least $1/2$. ∎

Let $B_1, B_2, \ldots, B_{2n}$ be these $2n$ bundles. By applying Lemma 2.6 we have

$$\sum_{1 \leq j \leq 2n} C_{\bar{v}}^{\mathcal{F}}(B_j) \leq \sum_{1 \leq i \leq n} \bar{v}_i(\mathcal{F}) = Z.$$

Note that here $C_{\bar{v}}^{\mathcal{F}}$ refers to the contribution with respect to $\bar{v}_i$. Therefore, at least one of the bundles contributes less than $Z/(2n) < 1/4$ to $Z$. Let $B_k$ be one such bundle, i.e., $C_{\bar{v}}^{\mathcal{F}}(B_k) < 1/4$. Let $b_{kj}$ be the fraction of item $j$ belonging to bundle $B_k$ and let $\mathcal{F}'$ be the allocation that defines the contribution of bundle $B_k$ to allocation $\mathcal{F}$ (see Program 5). Then $\sum_i f'_{ij} = \sum_i f_{ij} - b_{kj}$ for all $j$ and

$$\sum_{1 \leq i \leq n} \bar{v}_i(\mathcal{F}) - \sum_{1 \leq i \leq n} \bar{v}_i(\mathcal{F}') = C_{\bar{v}}^{\mathcal{F}}(B_k) < 1/4,$$

which means

$$\sum_{1 \leq i \leq n} \bar{v}_i(\mathcal{F}') > Z - \frac{1}{4}.$$

We now assign the items in $B_k$ to agent $i$, i.e., we consider the fractional allocation $\mathcal{F}''$ equal to $\mathcal{F}'$, except that for agent $i$, we have

$$f''_{i,j} = f'_{i,j} + b_{k,j} \qquad \text{for all } j \in [1 \ldots m]$$

Since the value of bundle $B_k$ to agent $a$ is at least $1/2$, we have $\bar{v}_i(\mathcal{F}'') - \bar{v}_i(\mathcal{F}') > 1/4$, and further

$$\sum_{1 \leq i \leq n} \bar{v}_i(\mathcal{F}'') > \sum_{1 \leq i \leq n} \bar{v}_i(\mathcal{F}') + \frac{1}{4} > (Z - \frac{1}{4}) + \frac{1}{4} = Z = \sum_{1 \leq i \leq n} \bar{v}_i(\mathcal{F}). \tag{8}$$

However, Inequality (8) contradicts the fact that allocation $\mathcal{F}$ maximizes the social welfare. Hence, $\mathcal{F}$ guarantees at least $1/4$ to all the agents.

Finally, let $\mathcal{R}$ be the randomized allocation obtained from $\mathcal{F}$ through (4). Then $v_i(\mathcal{R}) \geq v_i(\mathcal{F}) \geq \bar{v}_i(\mathcal{F}) \geq 1/4$ by Lemma 2.5. Thus, $\mathcal{R}$ is $1/4$-MMS ex-ante. This completes the proof. □

We remark that though we constructed a $1/4$-MMS allocation, we have no guarantee on the ex-post fairness of our allocation. In the next section, our goal is to improve this allocation to also guarantee a fraction of maximin-share ex-post.

# 4 Ex-ante and Ex-post Guarantees

Unfortunately, the randomized allocation obtained by Theorem 1.1 has no ex-post fairness guarantee. The issue is that we use Theorem 2.9 to convert the fractional allocation into a randomized one. However, Theorem 2.9 only guarantees that the ex-post value of each agent is at least the value of her fractional allocation minus the value of the heaviest item which is partially (and not fully) allocated to her in the fractional allocation. However, currently, we have no upper bound on the value of the allocated items, and therefore, the ex-post value of an agent might be close to 0. To resolve this, we perform two improvements on our allocation.

First, we allocate valuable items beforehand to keep the value of the remaining items as small as possible. We start by using a simple and very practical fact that is frequently used in previous studies [24, 32, 10, 5]: allocating one item to one agent and removing them from the instance does not decrease the maximin-share value of the remaining agents for the remaining items.

**Lemma 4.1.** *Removing one item and one agent from the instance does not decrease the maximin-share value of the remaining agents for the remaining items.*

Given that our goal is to construct a randomized allocation which is 1/4-MMS ex-ante, by Lemma 4.1 we can assume without loss of generality that the value of each item to each agent is less than 1/4; otherwise, we can reduce the problem using Lemma 4.1. However, a combination of this assumption and Theorem 1.1 still gives no ex-post guarantee: the ex-ante guarantee obtained by Theorem 1.1 is 1/4-MMS and assuming that the value of each item to each agent is less than 1/4 implies no lower-bound better than 0 on the ex-post MMS guarantee. To improve the ex-post guarantee, we revisit the proof of Theorem 2.9 and show that for our setting, a stronger guarantee can be achieved using the matching method for converting a fractional allocation into a randomized one. Indeed, we show that we can find a fractional allocation with a special structure that makes the transformation step more efficient. These ideas together help us achieve a randomized allocation with 1/4-MMS guarantee ex-ante and 1/8-MMS guarantee ex-post.

**Theorem 1.2.** *For any instance with* XOS *valuations, Algorithm 1 returns a randomized allocation that is* 1/4-MMS *ex-ante and* 1/8-MMS *ex-post.*

In the rest of this section, we prove Theorem 1.2. The algorithm we use for proving Theorem 1.2 is as follows:

(i). While there exists an item $b_j$ with value at least 1/4 to an agent $i$, allocate $b_j$ to agent $i$ and remove $i$ and $b_j$ respectively from $\mathcal{N}$ and $\mathcal{M}$.

(ii). Assuming $[n]$ is the set of the remaining agents, let $\mathcal{F}$ be an optimal solution of the following linear program.

$$
\begin{aligned}
\text{maximize} \quad & \sum_{1 \leq i \leq n} u_i \\
\text{subject to} \quad & \sum_{1 \leq i \leq n} f_{i,j} = 1 && \text{for all } j \\
& f_{i,j} \in \{0, 1/2, 1\} && \text{for all } i \text{ and } j \\
& u_i = \min(\frac{1}{2}, \max_k \sum_j u_{i,k}(b_j) f_{i,j}). && \text{for all } i \quad (9)
\end{aligned}
$$

(iii). Convert $\mathcal{F}$ into a randomized allocation using Lemma 4.3.

See Algorithm 1 for the pseudocode. Recall that by Lemma 4.1, after Step (i), the maximin-share value of the remaining agents for the remaining items is at least 1. For simplicity, we scale the valuations after the first step so that the MMS value of each remaining agent after the first step is exactly equal to 1. All agents $i$ who are allocated an item $b_j$ in Step (i), are also allocated $b_j$ in the fractional allocation. Thus, 1/4-MMS is guaranteed for $i$ in the final randomized allocation both ex-ante and ex-post. Hence, it is without loss of generality to ignore these agents and assume $n$ is the number of remaining agents after Step (i).

An equivalent description for Step (ii) is the following. For every agent $i$, define $\bar{v}_i$ as follows:

$$\forall S \subseteq \mathcal{M} \quad \bar{v}_i(S) = \min(1/2, v_i(S)).$$

**Algorithm 1** ExPostExAnteMMS$(\mathcal{N}, \mathcal{M}, \mathcal{V})$

**Input:** Instance $(\mathcal{N}, \mathcal{M}, \mathcal{V})$.

**Output:** Allocation $\mathcal{R}$.

---

1: **while** there exists $b_j \in \mathcal{M}$ and $i \in \mathcal{N}$ s.t. $v_i(b_j) \geq 1/4$ **do** $\qquad\qquad$ ▷ Step (i)
2: $\quad\quad \mathcal{R}_i \leftarrow \{b_j\}$
3: $\quad\quad \mathcal{M} \leftarrow \mathcal{M} \setminus \{b_j\}$
4: $\quad\quad \mathcal{N} \leftarrow \mathcal{N} \setminus \{i\}$
5: Let $\bar{v}_i(\cdot) = \min(1/2, v_i(\cdot))$
6: Let $\Pi$ be the set of all half-integral allocations of $\mathcal{M}$ to $\mathcal{N}$
7: Let $\mathcal{F} = \arg\max_{F \in \Pi} \sum_{i \in \mathcal{N}} \bar{v}_i(F_i)$ $\qquad\qquad\qquad\qquad\qquad\qquad$ ▷ Step (ii)
8: Let $\mathcal{R}$ be the randomized allocation obtained from $\mathcal{F}$ by Lemma 4.3. $\qquad$ ▷ Step (iii)
9: Return $\mathcal{R}$

---

Let $\bar{v} = (\bar{v}_1, \ldots, \bar{v}_n)$ and return a half-integral allocation $\mathcal{F}$ that maximizes social welfare with respect to $\bar{v}$, i.e., $\mathcal{F} = \arg\max_{A \in \Pi} \sum_{i \in \mathcal{N}'} \bar{v}_i(A_i)$ where $\Pi$ is the set of all half-integral allocations of $\mathcal{M}$ to $\mathcal{N}$. The goal in Step (ii) is to find a fractional allocation that is 1/4-MMS. However, we want this allocation to have a special structure that facilitates constructing the randomized allocation. Therefore, instead of directly choosing the allocation that maximizes social welfare, we consider $\bar{v}_i$ as the valuation function of agent $i$ and return a half-integral allocation. First we prove that $\mathcal{F}$ is 1/4-MMS. Otherwise, let $i$ be an agent that has a value less than 1/4 for her share. By Claim 3.5, we know that agent $i$ can distribute all the items (that have remained after Step (i)) into $2n$ bundles each with value at least 1/2 to her. Here, we construct these $2n$ bundles more carefully.

Indeed, for every bundle in the optimal partitioning of agent $i$, we construct two bundles with a value of at least 1/2 as follows: we divide each item into two half-unit items and put each half-unit into one bundle. That way, for all items $b_j$, there are two bundles each of which contains one half of $b_j$.

Using the same deduction as we used in the proof of Theorem 1.2, we can say that since the number of remaining agents after Step (i) is $n$, the value of one agent for her bundle is less than 1/4, and the value of the rest of the agents for their bundles is at most 1/2, the social welfare of the allocation is less than $n/2$. Therefore, at least one of these $2n$ bundles, say $B_k$ contributes less than 1/4 to social welfare. Now we take back these items from other agents and allocate them to agent $i$. The reallocation increases social welfare as shown in the proof of Theorem 1.1. Note that also in this new allocation for every agent $i$ and item $b_j$, we have $f_{i,j} \in \{0, 1/2, 1\}$ which is a contradiction with the choice of $\mathcal{F}$.

**Observation 4.2.** Let $\mathcal{F}$ be the allocation after Step (ii). Then, for every agent $i$ we have $v_i(\mathcal{F}) \geq 1/4$. Furthermore, for every item $b_j$, we have $f_{i,j} \in \{0, 1/2, 1\}$.

**Proof.** Towards a contradiction assume $v_i(\mathcal{F}) < 1/4$ for some agent $i$. Let $B_1, B_2, \ldots, B_{2n}$ be the result of halving the bundles in the optimal partitioning of agent $i$, i.e., dividing each item into two half-unit items and putting each half-unit into one bundle. Let $\mathcal{F}'$ be the allocation (see Program 5) that defines the contribution of bundle $B_k$ to allocation $\mathcal{F}$. Then $\sum_i f'_{ij} = \sum_i f_{ij} - b_{kj}$ for all $j$ and $\sum_{1 \leq i \leq n} \bar{v}_i(\mathcal{F}) - \sum_{1 \leq i \leq n} \bar{v}_i(\mathcal{F}') = C_{\bar{v}}^{\mathcal{F}}(B_k) < 1/4$. We now assign the items in $B_k$ to agent $i$, i.e., we consider the fractional allocation $\mathcal{F}''$ equal to $\mathcal{F}'$, except that for agent $i$, we have

$$\forall_{1 \leq j \leq m} \qquad f''_{i,j} = f'_{i,j} + b_{k,j}.$$

Since the value of bundle $B_k$ to agent $a$ is at least 1/2, we have $\bar{v}_i(\mathcal{F}'') - \bar{v}_i(\mathcal{F}') > 1/4$, and hence $\sum_{1 \leq i \leq n} \bar{v}_i(\mathcal{F}'') > \sum_{1 \leq i \leq n} \bar{v}_i(\mathcal{F})$. Since for every agent $i'$ and item $b_j$, we have $f''_{i',j} \in \{0, 1/2, 1\}$ and hence a contradiction to the optimality of $\mathcal{F}$. $\qquad\square$

Now, we show how to convert $\mathcal{F}$ into a randomized allocation. Recall that the result of Theorem 2.9 does not provide us with an ex-post guarantee better than 0. Here, we give a more accurate analysis to prove that the outcome of our algorithm is 1/8-MMS. Our construction is based on the Birkhoff—von Neumann theorem: Every fractional perfect matching can be written as a linear combination of integral perfect matchings. We adopt the construction to our setting and, in particular, exploit the fact that all $f_{ij}$ are half-integral.

**Lemma 4.3.** *Assume that the valuations are additive and let $\mathcal{F}$ be a complete fractional allocation with $f_{ij} \in \{0, 1/2, 1\}$ for all $i$ and $j$. Then there is a randomized allocation $\mathcal{R}$ with $D(\mathcal{R}) = \{\mathcal{A}^1, \mathcal{A}^2\}$, such that*

- *For every agent $i$ we have $v_i(\mathcal{R}) = v_i(\mathcal{F})$.*

- *For every agent $i$ we have*

$$\min\left\{v_i(\mathcal{A}_i^1), v_i(\mathcal{A}_i^2)\right\} \geq v_i(\mathcal{R}) - \frac{\max\{v_i(b_j) \mid f_{ij} = 1/2\}}{2}.$$

**Proof.**    For each agent $i$, let $f_i = \sum_j f_{ij}$. Since $\sum_i f_i = m$, the number of agents with non-integral $f_i$ is even. We pair the agents with non-integral $f_i$ arbitrarily. For each pair, we create a new dummy item with a value of zero for all the agents and assign one-half of the dummy item to each agent in the pair. In this way, for every agent $i$, $f_i$ becomes an integer. Therefore, for the rest of the proof we assume that for every agent $i$, $f_i$ is an integer.

We now construct allocations $\mathcal{A}^1$ and $\mathcal{A}^2$ such that $f_{ij}$ is equal to the fraction of allocations in $\mathcal{R}$ that allocate item $b_j$ to $i$, i.e., if $f_{ij} = 1$ we allocate $b_j$ to $i$ in both allocations, if $f_{ij} = 0$, we allocate $b_j$ to $i$ in neither allocations, and if $f_{ij} = 1/2$ we allocate $b_j$ to $i$ in exactly one of the two allocations. For brevity, we define $\mathcal{M}_{1/2}$ and $\mathcal{N}_{1/2}$ as follows:

$$\mathcal{M}_{1/2} = \{b_j | \exists i : f_{i,j} = 1/2\}$$
$$\mathcal{N}_{1/2} = \{i | \exists b_j : f_{i,j} = 1/2\}$$

Consider a bipartite graph $G(X, Y)$ with parts $X$ and $Y$ as follows: For every item $b_j \in \mathcal{M}_{1/2}$ there is a vertex $y_j$ in $Y$ corresponding to item $b_j$. For every agent $i \in \mathcal{N}_{1/2}$, we have vertices $x_i^1, x_i^2, \ldots, x_i^{t_i}$ in $X$, where $t_i$ is half the number of items $b_j$ such that $f_{ij} = 1/2$, that is

$$t_i = \frac{\left|\{b_j | f_{i,j} = 1/2\}\right|}{2}.$$

Also, we add the following edges to $G$. For every $i \in \mathcal{N}_{1/2}$, order the items $b_j$ with $f_{ij} = 1/2$ in decreasing order of their value to agent $i$. Then we connect $x_i^1$ to the first two items, $x_i^2$ to the items with ranks three and four, and so on. In this way, all vertices in $X$ and $Y$ have degree two. Hence, $G$ decomposes into vertex disjoint cycles. Each cycle decomposes into two matchings (note that since the graph is bipartite, all the cycles have even length), and thus $G$ decomposes into two perfect matchings, say $M^1$ and $M^2$. We define allocations $\mathcal{A}^1$ and $\mathcal{A}^2$ as follows: for every item $b_j$, agent $i$, and $r \in \{1, 2\}$, we allocate item $b_j$ to agent $i$ in $\mathcal{A}^r$, if and only if either $f_{i,j} = 1$, or $M^r(y_j) = x_i^k$ for some $1 \leq k \leq t_i$, where $M^r(y_j)$ refers to the vertex matched with $y_j$ in $M^r$. Define $\mathcal{R}$ as a randomized allocation that selects $\mathcal{A}^1$ or $\mathcal{A}^2$, each with probability 1/2.

It is easy to check that for every agent $i$, $v_i(\mathcal{R}) = v_i(\mathcal{F})$. Here, we focus on the ex-post guarantee of $\mathcal{R}$. Fix an agent $i$ and Let $b_1$, $b_2$, $\ldots$, $b_{2t_i}$ be the items half-owned by $i$ in order of decreasing value for agent $i$. Then, by the way we construct $M^1$ and $M^2$, for all $1 \leq \ell \leq t_i$, $b_{2\ell-1}$ and $b_{2\ell}$

are allocated to $i$ in different allocations. Hence, the value of the $i^{\text{th}}$ bundle in either allocation $\mathcal{A}^r$ satisfies

$$v_i(\mathcal{A}_i^r) \geq v_i(b_2) + v_i(b_4) + \ldots + v_i(b_{2t_i}) + \sum_{j:f_{i,j}=1} v_i(b_j).$$

We can now bound $v_i(\mathcal{F}) - v_i(\mathcal{A}_i^r)$ from above.

$$
\begin{aligned}
v_i(\mathcal{F}) - v_i(\mathcal{A}_i^r) &\leq \frac{1}{2}\left(\sum_{1 \leq \ell \leq 2t_i} v_i(b_\ell)\right) - \sum_{1 \leq \ell \leq t_i} v_i(b_{2\ell}) \\
&= \frac{v_i(b_1)}{2} + \sum_{1 \leq \ell < t_i}\left(\frac{v_i(b_{2\ell}) + v_i(b_{2\ell+1})}{2} - v_i(b_{2\ell})\right) + \frac{v_i(b_{2t_i})}{2} - v_i(b_{2t_i}) \\
&\leq \frac{v_i(b_1)}{2}.
\end{aligned}
$$

$\square$

Now we are ready to prove Theorem 1.2.

**Theorem 1.2.** *For any instance with* XOS *valuations, Algorithm 1 returns a randomized allocation that is* $1/4$-MMS *ex-ante and* $1/8$-MMS *ex-post.*

**Proof.** The ex-ante guarantee follows from Observation 4.2 and Lemma 4.

Let $\mathcal{A}^1$ and $\mathcal{A}^2$ be the integral allocations obtained by Lemma 4.3. Consider any agent $i$, and let $u_{i,i'}$ be such that $v_i(\mathcal{R}) = \sum_j f_{ij} u_{i,i'}(b_j)$. Then, by Lemma 4.3 for $r \in \{1,2\}$ we have

$$u_{i,i'}(\mathcal{A}_i^r) \geq u_{i,i'}(\mathcal{R}_i^\ell) - \frac{\max\{u_{i,i'}(b_j) \mid f_{i,j}=1/2\}}{2}$$

and since by Lemma 4.1 we know the value of each item for each agent is less than $1/4$, we have

$$v_i(\mathcal{A}_i^\ell) \geq v_i(\mathcal{R}_i) - \frac{\max_j v_i(b_j)}{2} > \frac{1}{4} - \frac{1}{8} = \frac{1}{8}.$$

Hence, the ex-post guarantee holds as well. $\square$

# 5   3/13-MMS Allocation

In this section, we improve the best approximation guarantee of MMS for deterministic allocations in the fractionally subadditive setting. We show that a factor $3/13 \approx 0.230769$ of the maximin-share of every agent is possible. Before this work, the best approximation guarantee for maximin-share in the XOS setting was 0.219225-MMS [32].

Our algorithm for improving the ex-post guarantee is based on our previous algorithms plus two additional steps and a more in-depth analysis. In this algorithm, before finding the allocation that maximizes social welfare, we strengthen our upper bound on the value of items. For this, we add two more steps to our algorithm in which we satisfy some of the agents with two items and three items. In contrast to the first step (i.e., allocating single items to agents), these steps might decrease the maximin-share value of the remaining agents for the remaining items. Let $t = 6/13$. The goal is to find a $t/2$-MMS allocation. Our allocation algorithm is as follows:

(i). While there exists an item $b_j$ with value at least $t/2$ to an agent $i$, allocate $b_j$ to agent $i$ and remove $i$ and $b_j$ respectively from $\mathcal{N}$ and $\mathcal{M}$.

**Algorithm 2** approxMMS($\mathcal{N}, \mathcal{M}, \mathcal{V}$)
**Input:** Instance $(\mathcal{N}, \mathcal{M}, \mathcal{V})$.
**Output:** Allocation $\mathcal{A}$.

---

1: Let $t = 6/13$
2: **while** there exists $b_j \in \mathcal{M}$ and $i \in \mathcal{N}$ s.t. $v_i(b_j) \geq t/2$ **do** $\qquad\qquad$ ▷ Step 1
3: $\qquad \mathcal{A}_i \leftarrow \{b_j\}$
4: $\qquad \mathcal{M} \leftarrow \mathcal{M} \setminus \{b_j\}$
5: $\qquad \mathcal{N} \leftarrow \mathcal{N} \setminus \{i\}$
6: **while** there exists $b_j, b_k \in \mathcal{M}$ and $i \in \mathcal{N}$ s.t. $v_i(\{b_j, b_k\}) \geq t/2$ **do** $\qquad$ ▷ Step 2
7: $\qquad \mathcal{A}_i \leftarrow \{b_j, b_k\}$
8: $\qquad \mathcal{M} \leftarrow \mathcal{M} \setminus \{b_j, b_k\}$
9: $\qquad \mathcal{N} \leftarrow \mathcal{N} \setminus \{i\}$
10: **while** there exists $b_j, b_k, b_s \in \mathcal{M}$ and $i \in \mathcal{N}$ s.t. $v_i(\{b_j, b_k, b_s\}) \geq t/2$ **do** $\quad$ ▷ Step 3
11: $\qquad \mathcal{A}_i \leftarrow \{b_j, b_k, b_s\}$
12: $\qquad \mathcal{M} \leftarrow \mathcal{M} \setminus \{b_j, b_k, b_s\}$
13: $\qquad \mathcal{N} \leftarrow \mathcal{N} \setminus \{i\}$
14: Let $\mathcal{N}' = \mathcal{N}$, $\mathcal{M}' = \mathcal{M}$ and $\bar{v}_i(\cdot) = \min(t, v_i(\cdot))$
15: Let $\Pi$ be the set of all allocations of $\mathcal{M}'$ to $\mathcal{N}'$
16: Let $\mathcal{A} = \arg\max_{A \in \Pi} \sum_{i \in \mathcal{N}'} \bar{v}_i(A_i)$ $\qquad\qquad\qquad\qquad\qquad\qquad\qquad$ ▷ Step 4
17: Return $\mathcal{A}$

---

(ii). While there exists a pair of items $b_j, b_k$ with total value of at least $t/2$ to some agent $i$, allocate $\{b_j, b_k\}$ to agent $i$, remove both goods from $\mathcal{M}$, and remove agent $i$ from $\mathcal{N}$.

(iii). While there exists a triple of items $b_j, b_k, b_s$ with total value of at least $t/2$ to some agent $i$, allocate $\{b_j, b_k, b_s\}$ to agent $i$, remove all three goods from $\mathcal{M}$, and remove agent $i$ from $\mathcal{N}$.

(iv). For the remaining agents $\mathcal{N}'$ and items $\mathcal{M}'$, proceed as follows: for every agent $i$, define $\bar{v}_i$ as follows:
$$\forall S \subseteq \mathcal{M} \quad \bar{v}_i(S) = \min(t, v_i(S)).$$

Let $\bar{v} = (\bar{v}_1, \ldots, \bar{v}_n)$ and return an allocation $\mathcal{A}$ that maximizes social welfare with respect to $\bar{v}$, i.e., $\mathcal{A} = \arg\max_{A \in \Pi} \sum_{i \in \mathcal{N}'} \bar{v}_i(A_i)$ where $\Pi$ is the set of all allocations of $\mathcal{M}'$ to $\mathcal{N}'$.

In the rest of this section, we analyze the above algorithm. See Algorithm 2 for the pseudocode. By Lemma 4.1, after Step (i), the MMS value of all the agents is at least 1. Let $n$ be the number of remaining agents after Step (i). We denote by $n_1$ and $n_2$, the number of agents that are satisfied in Steps (ii) and (iii) respectively and let $n' = n - n_1 - n_2 = |\mathcal{N}'|$ be the number of remaining agents after Step (iii). In contrast to the first step, Step (ii) and (iii) might decrease the maximin-share value of the remaining agents for the remaining items. However, we prove that the remaining items satisfy certain special structural properties.

**Observation 5.1.** Since no item can satisfy any remaining agent after Step (i), for every agent $i$ and every item $b_j$, we have $v_i(\{b_j\}) < t/2$.

Also, by the method that we allocate the items in Step (iii), after this step the following observation holds.

**Observation 5.2.** Since after Step (iii), no triple of items can satisfy an agent, for every different items $b_j, b_k, b_s$ and every agent $i$ we have $v_i(\{b_j, b_k, b_s\}) < t/2$.

Note that since the valuations are XOS, Observation 5.2 implies no upper bound better than $t/2$ on the value of a single item to an agent. For example, consider the following extreme scenario: for a small constant $\epsilon > 0$, the value of every non-empty subset of items to agent $i$ is equal to $t/2 - \epsilon$. It is easy to check that this valuation function is XOS. For this case, the value of every triple of items is also equal to $t/2 - \epsilon$, but this implies no upper bound better than $t/2$ on the value of a single item.

**Lemma 5.3.** *Fix a remaining agent $i$ and consider the $n$ bundles with value at least $1$ in an MMS partition of agent $i$ after Step (i). Put these bundles into 4 different sets $B_0, B_1, B_2, B_{\geq 3}$, where for $0 \leq \ell \leq 2$, set $B_\ell$ contains bundles that lose exactly $\ell$ items in Steps (ii) and (iii), and $B_{\geq 3}$ contains bundles that lose at least three items in these steps. After Step (iii), the following inequality holds:*

$$n' \leq |B_0| + \frac{2}{3}|B_1| + \frac{1}{3}|B_2|.$$

**Proof.** Since each satisfied agent in step $k$ receives $k$ items, we have:

$$2n_1 + 3n_2 \geq |B_1| + 2|B_2| + 3|B_{\geq 3}|.$$

Thus,

$$n_1 + n_2 \geq \frac{2}{3}n_1 + n_2 \geq \frac{1}{3}|B_1| + \frac{2}{3}|B_2| + |B_{\geq 3}|, \tag{10}$$

and therefore,

$$
\begin{aligned}
n' &= n - n_1 - n_2 \\
&\leq n - \frac{1}{3}|B_1| - \frac{2}{3}|B_2| - |B_{\geq 3}| & \text{(Inequality 10)} \\
&= |B_0| + \frac{2}{3}|B_1| + \frac{1}{3}|B_2|. & (n = |B_0| + |B_1| + |B_2| + |B_{\geq 3}|)
\end{aligned}
$$

$\square$

Finally, in Step (iv), we find the integral allocation $\mathcal{A}$ that maximizes social welfare with respect to $\bar{v}$ for the remaining agents. Let

$$Z = \sum_{i \in \mathcal{N}'} \bar{v}_i(\mathcal{A}_i).$$

Since for each remaining agent $i$, $\bar{v}_i(\mathcal{A}_i)$ is upper-bounded by $t$, we have $Z \leq n't$. If for every agent $i$, $v_i(\mathcal{A}_i) \geq t/2$ holds, then $\mathcal{A}$ is $t/2$-MMS, and we are done. Therefore, for the rest of this section, assume that for an agent $i^*$, we have $v_{i^*}(\mathcal{A}_{i^*}) < t/2$.

**Lemma 5.4.** *For all sets $S \subseteq M$, $C_{\bar{v}}^{\mathcal{A}}(S) \geq \bar{v}_{i^*}(S) - \bar{v}_{i^*}(\mathcal{A}_{i^*})$.*

**Proof.** Let allocation $\mathcal{A}'$ be as following. For all agents $i$, $\mathcal{A}'_i = \mathcal{A}_i \setminus S$. Basically, $\mathcal{A}'$ is allocation $\mathcal{A}$ after removing all the items in $S$ from the bundles they belong to. We have

$$\sum_{i \in \mathcal{N}} \bar{v}_i(\mathcal{A}'_i) = \sum_{i \in \mathcal{N}} \bar{v}_i(\mathcal{A}_i) - C_{\bar{v}}^{\mathcal{A}}(S). \tag{11}$$

Now let $\mathcal{A}''$ be allocation $\mathcal{A}'$ after allocating $S$ to agent $i^*$. I.e., for all agents $i \neq i^*$, $\mathcal{A}''_i = \mathcal{A}'_i$ and $\mathcal{A}''_{i^*} = \mathcal{A}'_{i^*} \cup S$. We have

$$
\begin{aligned}
\sum_{i \in \mathcal{N}} \bar{v}_i(\mathcal{A}_i) &\geq \sum_{i \in \mathcal{N}} \bar{v}_i(\mathcal{A}''_i) && (\mathcal{A} = \text{argmax}_{A \in \Pi} \textstyle\sum_{i \in \mathcal{N}'} \bar{v}_i(A_i)) \\
&= \sum_{i \in \mathcal{N} \setminus \{i^*\}} \bar{v}_i(\mathcal{A}'_i) + \bar{v}_{i^*}(\mathcal{A}'_{i^*} \cup S) \\
&= \left( \sum_{i \in \mathcal{N}} \bar{v}_i(\mathcal{A}_i) - C^{\mathcal{A}}_{\bar{v}}(S) - \bar{v}_{i^*}(\mathcal{A}'_{i^*}) \right) + \bar{v}_{i^*}(\mathcal{A}'_{i^*} \cup S) && (\text{Inequality (11)}) \\
&\geq \sum_{i \in \mathcal{N}} \bar{v}_i(\mathcal{A}_i) - C^{\mathcal{A}}_{\bar{v}}(S) - \bar{v}_{i^*}(\mathcal{A}'_{i^*}) + \bar{v}_{i^*}(S). && (\bar{v}_{i^*}(\mathcal{A}'_{i^*} \cup S) \geq \bar{v}_{i^*}(S))
\end{aligned}
$$

Therefore, $C^{\mathcal{A}}_{\bar{v}}(S) \geq \bar{v}_{i^*}(S) - \bar{v}_{i^*}(\mathcal{A}_{i^*})$. $\qquad\square$

Let $B_0, B_1$, and $B_2$ be the sets defined for agent $i^*$ in Lemma 5.3. In Lemmas 5.5, 5.6 and 5.7, we give lower bounds on the contribution of the bundles in $B_0$, $B_1$ and $B_2$ to $\mathcal{A}$ respectively.

**Lemma 5.5.** *After Step (iii), for all bundles $X \in B_0$, there exists a partition of $X$ into $X^1$ and $X^2$ such that $C^{\mathcal{A}}_{\bar{v}}(X^1) + C^{\mathcal{A}}_{\bar{v}}(X^2) \geq t$.*

**Proof.** The idea is to partition the set $X$ into two bundles $X^1$ and $X^2$ each with value at least $t$ to agent $i^*$. Then using Lemma 5.4, we prove the contribution of each of these bundles to $\mathcal{A}$ is at least $t/2$ and thus the total contribution is at least $t$.

For a fixed bundle $X \in B_0$, let $j$ be such that $u_{i^*,j}(X) = v_{i^*}(X) \geq 1$. Let $g_1$ and $g_2$ be two different most valuable items in $X$ with respect to $u_{i^*,j}$, i.e., for all items $g \in X \setminus \{g_1, g_2\}$, $u_{i^*,j}(g_1) \geq u_{i^*,j}(g_2) \geq u_{i^*,j}(g)$. Let $X^1$ be a minimal subset of $X$ such that $\{g_1, g_2\} \subset X^1$ and $u_{i^*,j}(X^1) \geq t$. Let $X^2 = X \setminus X^1$. Since $X^1$ is minimal, for all $g \in X_1$, $u_{i^*,j}(X^1 \setminus \{g\}) < t$. Also, by Observation 5.2, for all $g \in X^1 \setminus \{g_1, g_2\}$, $u_{i^*,j}(\{g_1, g_2, g\}) \leq v_{i^*}(\{g_1, g_2, g\}) < t/2$ and thus, $u_{i^*,j}(g) < t/6$. Therefore, for all $g \in X^1 \setminus \{g_1, g_2\}$,

$$
\begin{aligned}
u_{i^*,j}(X^2) &\geq 1 - u_{i^*,j}(X^1) && (u_{i^*,j}(X^1 \cup X^2) \geq 1) \\
&= 1 - \left( u_{i^*,j}(X^1 \setminus \{g\}) + u_{i^*,j}(g) \right) && (\text{by additivity of } u_{i^*,j}) \\
&> 1 - \frac{7}{6}t \\
&= t. && (t = 6/13)
\end{aligned}
$$

Hence, we have $v_{i^*}(X^1) \geq u_{i^*,j}(X_1) \geq t$ and $v_{i^*}(X^2) \geq u_{i^*,j}(X_2) \geq t$. Now by Lemma 5.4, we have

$$
\begin{aligned}
C^{\mathcal{A}}_{\bar{v}}(X^1) + C^{\mathcal{A}}_{\bar{v}}(X^2) &\geq \left( \bar{v}_{i^*}(X^1) - \bar{v}_{i^*}(\mathcal{A}_{i^*}) \right) + \left( \bar{v}_{i^*}(X^2) - \bar{v}_{i^*}(\mathcal{A}_{i^*}) \right) \\
&> 2(t - \frac{1}{2}t) = t.
\end{aligned}
$$

$\qquad\square$

**Lemma 5.6.** *After Step (iii), for all bundles $X \in B_1$, there exists a partition of $X$ into $X^1$ and $X^2$ such that $C^{\mathcal{A}}_{\bar{v}}(X^1) + C^{\mathcal{A}}_{\bar{v}}(X^2) \geq \frac{2}{3}t$.*

**Proof.** Fix a bundle $X \in B_1$. By Lemma 4.1, the MMS value of agent $i^*$ is at least 1 after Step (i). By Observation 5.1, $v_{i^*}(g) < t/2$ for all remaining items $g$ after Step (i). Since $X$ is a bundle in an MMS partition of agent $i^*$ after Step (i) and after the removal of one item $g$, we have

$$
v_{i^*}(X) > 1 - t/2. \tag{12}
$$

Let $j$ be such that $u_{i^*,j}(X) = v_{i^*}(X)$. Let $g_1$ and $g_2$ be two different most valuable items in $X$ with respect to $u_{i^*,j}$, i.e., for all items $g \in X \setminus \{g_1, g_2\}$, $u_{i^*,j}(g_1) \geq u_{i^*,j}(g_2) \geq u_{i^*,j}(g)$. Let $X^1$ be a minimal subset of $X$ such that $\{g_1, g_2\} \subset X^1$ and $u_{i^*,j}(X^1) \geq 2t/3$. Let $X^2 = X \setminus X^1$. Since $X^1$ is minimal, for all $g \in X_1$, $u_{i^*,j}(X^1 \setminus \{g\}) < 2t/3$. Also, by Observation 5.2, for all $g \in X^1 \setminus \{g_1, g_2\}$, $u_{i^*,j}(\{g_1, g_2, g\}) \leq v_{i^*}(\{g_1, g_2, g\}) < t/2$ and thus, $u_{i^*,j}(g) < t/6$. Therefore,

$$u_{i^*,j}(X_1) = u_{i^*,j}(X_1 \setminus \{g\}) + u_{i^*,j}(g) \qquad \text{(by additivity of } u_{i^*,j})$$
$$< \frac{2}{3}t + \frac{1}{6}t = \frac{5}{6}t.$$

Therefore, for all $g \in X^1 \setminus \{g_1, g_2\}$, we have

$$C_{\bar{v}}^{\mathcal{A}}(X^1) + C_{\bar{v}}^{\mathcal{A}}(X^2) \geq \left(\bar{v}_{i^*}(X^1) - \bar{v}_{i^*}(\mathcal{A}_{i^*})\right) + \left(\bar{v}_{i^*}(X^2) - \bar{v}_{i^*}(\mathcal{A}_{i^*})\right) \qquad \text{(Lemma 5.4)}$$
$$= \left(\min(t, v_{i^*}(X^1)) - \bar{v}_{i^*}(\mathcal{A}_{i^*})\right) + \left(\min(t, v_{i^*}(X^2)) - \bar{v}_{i^*}(\mathcal{A}_{i^*})\right)$$
$$> \left(\min(t, u_{i^*,j}(X^1)) - \frac{1}{2}t\right) + \left(\min(t, u_{i^*,j}(X^2)) - \frac{1}{2}t\right)$$
$$\geq u_{i^*,j}(X^1) + \min(t, 1 - \frac{1}{2}t - u_{i^*,j}(X^1)) - t \qquad (u_{i^*,j}(X^1) < 5t/6)$$
$$\geq \min(u_{i^*,j}(X^1), 1 - \frac{3}{2}t)$$
$$\geq \frac{2}{3}t.$$

$\square$

**Lemma 5.7.** *After Step (iii), for all bundles $X \in B_2$, $C_{\bar{v}}^{\mathcal{A}}(X) \geq \frac{1}{2}t$.*

**Proof.** Fix a bundle $X \in B_2$. By Lemma 4.1, the MMS value of agent $i^*$ is at least 1 after Step (i). By Observation 5.1, $v_{i^*}(\{g\}) < t/2$ for all remaining items $g$ after Step (i). Since $X$ is a bundle in an MMS partition of agent $i^*$ after Step (i) and after the removal of two items like $g$, we have $v_{i^*}(X) > 1 - t > t$. Therefore, $\bar{v}_{i^*}(X) = \min(t, v_{i^*}(X)) = t$. Now by Lemma 5.4,

$$C_{\bar{v}}^{\mathcal{A}}(X) \geq \bar{v}_{i^*}(X) - \bar{v}_{i^*}(\mathcal{A}_i) > t - \frac{1}{2}t = \frac{1}{2}t.$$

$\square$

**Theorem 1.3.** *For any instance with XOS valuations, Algorithm 2 returns a 3/13-MMS allocation.*

**Proof.** Let $\mathcal{A}$ be the output of Algorithm 2. Towards a contradiction, assume for agent $i^*$, $v_{i^*}(\mathcal{A}_{i^*}) < 3/13 = t/2$. For all agents $i$ which are removed during the first three steps, we have $v_i(\mathcal{A}_i) \geq t/2 = 3/13$. Therefore, $i^* \in \mathcal{N}'$. For all $X \in B_0$, let $X^1$ and $X^2$ be as defined in Lemmas 5.5 and 5.6. We have

$$t(n' - \frac{1}{2}) > \sum_{i \in \mathcal{N}'} \bar{v}_i(\mathcal{A}_i) \qquad \text{(for all } i \in \mathcal{N}', \bar{v}_i(\mathcal{A}_i) \leq t \text{ and } \bar{v}_{i^*}(\mathcal{A}_{i^*}) < t/2)$$
$$\geq \sum_{X \in B_0} \left(C_{\bar{v}}^{\mathcal{A}}(X^1) + C_{\bar{v}}^{\mathcal{A}}(X^2)\right) + \sum_{X \in B_1} \left(C_{\bar{v}}^{\mathcal{A}}(X^1) + C_{\bar{v}}^{\mathcal{A}}(X^2)\right) + \sum_{X \in B_2} C_{\bar{v}}^{\mathcal{A}}(X)$$
$$\text{(Lemma 2.6)}$$
$$\geq t|B_0| + \frac{2}{3}t|B_1| + \frac{1}{2}t|B_2| \qquad \text{(Lemmas 5.5, 5.6 and 5.7)}$$
$$\geq tn', \qquad \text{(Lemma 5.3)}$$

which is a contradiction. Therefore, such an agent $i^*$ does not exist and $\mathcal{A}$ is a 3/13-MMS allocation.

$\square$

# 6 Future Directions

We developed randomized and deterministic allocations guaranteeing approximations of maximin-share for fractionally subadditive valuations. For randomized allocations, we derived simultaneous ex-ante and ex-post guarantees. Several interesting questions remain open.

The most straight-forward direction is to improve approximation guarantees for both ex-ante and ex-post cases. The first result we obtained in this paper is an allocation that is simultaneously 1/4-MMS ex-ante and 1/8-MMS ex-post. Also, we proved the existence of an allocation which is 3/13-MMS ex-post. None of these results are known to be tight. Therefore, the following questions remain open.

**Question 1.** *Can we find a randomized allocation with ex-ante* MMS *approximation guarantee better than 1/4 for fractionally subadditive valuations?*

Note that the result of Lemma 3.3 shows that no randomized allocation can guarantee a fraction better than 3/4-MMS ex-ante. Therefore, a gap of $[1/4, 3/4)$ remains between the best upper bound and the best lower bound for guaranteeing MMS ex-ante for fractionally subadditive valuations.

**Question 2.** *Can we find an allocation with ex-post* MMS *approximation guarantee better than 3/13?*

Moreover, for simultaneous ex-ante and ex-post guarantees there might be room for improving Theorem 1.2.

**Question 3.** *Can we find better allocations guaranteeing* MMS *both ex-ante and ex-post simultaneously?*

We believe that extending the idea in the second step of the proof of Theorem 1.1 might help improving the ex-post guarantee: we can add additional steps to check if some agent can be satisfied with two, three or more items. This way, we would have better upper bounds on the value of remaining items that head to the next step. However, having such additional steps need a more careful analysis of the maximin-share value of the remaining agents.

Another notable point about the results of this paper is that the algorithms defined in Theorems 1.1, 1.2 and 1.3 are not necessarily implementable in polynomial time. The reason is that calculating the exact value of $\mathsf{MMS}_i$ for each agent and also calculating an MMS partition of the agents is not implementable in polynomial time. Currently, the best polynomial time ex-post guarantee for MMS in the XOS setting is 1/8 [24]. This result can also be considered as the best polynomial time algorithm for finding an allocation with ex-ante guarantee for MMS in the additive case.

**Question 4.** *Can we find an allocation with a constant ex-ante* MMS *approximation guarantee better than 1/8 that can be implemented in polynomial time?*

Also, an interesting open direction is to find similar results for other classes of valuation functions, including submodular and subadditive valuations.

**Question 5.** *Can we find simultaneous ex-ante and ex-post guarantees for* MMS *for submodular and subadditive set functions?*

Currently the best guarantee for the submodular setting is 10/27-MMS ex-post [34]. Also, for the subadditive case, the best known ex-post approximation guarantee is $O(\frac{1}{\log n \log \log n})$.