# OpenReview forum: "Randomized and Deterministic Maximin-share Approximations for Fractionally Subadditive Valuations"
_NeurIPS.cc/2023/Conference — NeurIPS 2023 poster_

### Official Review · Reviewer_U5pm · 2023-07-04

**Soundness:** 3 good
**Presentation:** 3 good
**Contribution:** 3 good
**Rating:** 7
**Confidence:** 4

**Summary:**

This paper considers several classical fair division problems. This paper considers both the deterministic and randomized allocations under the XOS valuations. The XOS valuation is a subclass of the subadditive functions and it generalizes the submodular function. The deterministic allocation is an assignment between a partition of the items and agents. Randomized allocation is a distribution defined over a batch of deterministic allocation. The expectation value of the randomized allocation is called ex-ante, while the ex-post ensures fairness after fixing some outcome. The fairness criteria studied in this paper are MMS and proportional share.

The main contribution of this work is the existence of a randomized allocation that is 1/4-MMS ex-ante and 1/8-MMS ex-post. They also show that any ex-ante MMS guarantee for randomized allocation is at most 3/4. This is the first known result for randomized allocation under the MMS criteria and the XOS function. Besides this, by extending the technique from the randomized allocation, they also prove that there exists a 3/13-MMS deterministic allocation under the XOS function. This improves the previous best-known result by 0.21.

**Strengths:**

1. I appreciated that the submission is carefully written and structured, so reads well given the technicality of the material. Especially, the flow of the paper is well-designed. The presentation of the algorithmic idea is also clear.

2. The studied problems are classical fair division problems and I also believe that the paper made a solid contribution to the fair division field.



**Weaknesses:**

I believe the paper is a quality submission and my evaluation is based on technical contribution. On the downside, I didn't see a clear connection between the studied problem and the machine-learning community. The studied problem is a pretty classical fair division problem and thus the paper seems more suitable for conferences like EC/WINE/AAAI/IJCAI.

**Questions:**

I don't have any specific questions.

**Limitations:**

This is a theoretical paper, there is no potential negative societal impact.

---

### Official Review · Reviewer_TogA · 2023-07-06

**Soundness:** 4 excellent
**Presentation:** 3 good
**Contribution:** 4 excellent
**Rating:** 7
**Confidence:** 4

**Summary:**

The paper studies fair allocation under fractionally subadditive (XOS) valuations, where a valuation function is XOS if it can be represented by a maximization among a set of additive valuation functions. Both deterministic allocation and randomized allocation settings are considered in the paper. For the deterministic allocation, they improve the previous bound to 3/13; while for the randomized allocation, they obtain an algorithm with 1/4-MMS ex-ante and 1/8-MMS ex-post, which means that for any agent, the expected value obtained from the allocation is at least 1/4-MMS and the worst-case value is at least 1/8-MMS. The negative side is also investigated. The authors show that any randomized allocation algorithm cannot have an ex-ante ratio better than 3/4.

**Strengths:**

The paper makes technical contributions. It improves the previous bound for the deterministic fair allocation under XOS valuations, and obtains the first results for the randomized allocation under the MMS criteria. These results may influence some future works on the fair division area.


**Weaknesses:**

It might be better to present detailed proof of at least one technical lemma in the paper, but I understand that the space is too limited.

**Questions:**

Considering the special case that the additive functions behind each agent's XOS valuations have a value of either 1 or 0 for each element, any interesting result here?

**Limitations:**

Yes, the authors state some limitations and future directions in the last section.

---

> ### Author Rebuttal · Authors · 2023-08-03
>
> That is an interesting special case to consider. After receiving your review, we found a negative result that exact $MMS$ allocations cannot be guaranteed in this case. For the positive results, to the best of our knowledge, no better approximation factor is know for this particular setting.

---

> > ### Comment · Reviewer_TogA · 2023-08-16
> >
> > I see. Thanks for the response.

---

### Official Review · Reviewer_DbzA · 2023-07-06

**Soundness:** 4 excellent
**Presentation:** 4 excellent
**Contribution:** 4 excellent
**Rating:** 7
**Confidence:** 3

**Summary:**

This paper focuses on the problem of fair allocation of indivisible items to agents with fractionally subadditive valuations. The authors propose both deterministic and randomized allocation algorithms with improved approximation guarantees for the maximin-share objective.

In the deterministic setting, the authors provide an algorithm that achieves an approximation guarantee of 3/13 \approx 0.230, which improves the best-known approximation in the literature. This algorithm also uses the idea of maximizing capped social welfare, but before this step, it carefully analyzed how to allocate large items.

In the randomized setting, the authors prove that there is a randomized allocation that is 1/4 MMS in expectation, and complement this result with an upper bound of 3/4. The authors also show how to decompose this randomized allocation to deterministic ones that are all 1/8 MMS, which means in the best-of-both-worlds setting, there is an allocation algorithm that achieves an approximation guarantee of 1/4 ex-ante and 1/8 ex-post.

**Strengths:**

1. Improved approximation guarantees for XOS valuations: The paper proposes a deterministic allocation algorithm with improved approximation guarantees for the maximin-share objective.

2. The best-of-both-worlds setting: The authors show the existence of a randomized allocation that is ¼ MMS in expectation and propose a new decomposition method to decompose the randomized allocation to 1/8 MMS deterministic allocations.

**Weaknesses:**

1.	The results are still not tight, but I do not view this as a drawback.

2.	In the best-of-both-worlds setting, the ex-post guarantee 1/4 is worse than the best-known deterministic guarantee. A better ex-post guarantee is expected. Again, I think this is not a drawback.

**Questions:**

How large is the support of the randomized allocation in the best-of-both-worlds setting (in terms of # realizations)?

Thanks for the authors' answer in the rebuttal.

---

> ### Author Rebuttal · Authors · 2023-08-03
>
> The size of the support is $2$. Due to the space limit, we could not discuss how we convert the fractional allocation into a randomized one. The idea is to construct two allocations such that for all items like $g$ which are fractionally allocated to two agents like $i$ and $j$, $g$ is allocated to $i$ in exactly one of the realizations and to $j$ in the other one.

---

> > ### Comment · Reviewer_DbzA · 2023-08-18
> >
> > Thank you for the response. I don't have other questions.

---

### Official Review · Reviewer_Y5Uh · 2023-07-06

**Soundness:** 3 good
**Presentation:** 2 fair
**Contribution:** 3 good
**Rating:** 6
**Confidence:** 4

**Summary:**

This paper studies the problem of fair division of indivisible goods for XOS valuations. The notion of fairness considered here is the well-studied maximin-share (MMS). While it is known that an MMS allocation may fail to exist, even for additive valuations, obtaining positive results for increasingly better approximations of MMS is an active area of research.

The paper proves the following two results for the setting of XOS valuations:
- Existence of a 3/13-MMS allocation. This improves over the previously best known 0.219225-MMS
- A "best-of-both-worlds" result: Existence of a distribution over allocations that is 1/4-MMS ex-ante, and 1/8-MMS ex-post.

**Strengths:**

- the problem and results should be of interest to the fair division community
- technically solid

**Weaknesses:**

- a bit unclear how new the techniques are, and how they compare to the ones in prior work
- the writing is a bit confusing at times

**Questions:**

Questions:

1. In the overview of prior work, it would be nice to mention which of these results from prior work also yield efficient algorithms and which are mainly existential. Could you please comment on this, e.g., regarding the results in table 1? Are they all inefficient algorithms?

2. It would be nice to include a brief discussion on what the obstacles are to further improve the approximation guarantees. Are there fundamental obstacles that would require new techniques, or could the same ideas be pushed further?


Comments:

- lines 15-20: It is a bit misleading to say that the "basic scenario" for fair division is the indivisible setting; this has only been the case in recent years. Traditionally the problem has been studied in the divisible setting "for decades".

- table 1: is there a typo in "log n n"? perhaps "n log n" would be clearer

- line 63: "envy-free" has not been introduced yet (even informally)

- line 66: briefly and informally define what is meant by "best-of-both-worlds" in this context

- line 68: "for a more general class of fractionally subadditive valuation functions": the previous sentence talks about subadditive valuations, so this sentence does not make sense

- line 73: "fractionally subadditive set functions are closely related to subaddditive set functions" What do you mean? Isn't the first just a subset of the second?

- line 165: what if u_{i,i'} is not unique?

- line 180: the normalization assumption MMS_i = 1 is something that cannot be achieved efficiently. Is this the only reason why your algorithms are not efficient? I think it would be nice to comment on the non-efficiency of the algorithms more generally in the introduction.

- line 195: Instead of "determine", perhaps "define" would be more appropriate here

- lines 246-247: the wording of Observation 3.2 is a bit confusing. It doesn't make sense to first say that there is an allocation and then that there is an instance. It should be the other way.

- Lemma 3.3: does your construction use any ideas from existing similar upper bounds?

- Example 3.4: the first sentence does not make sense, since the proof of Observation 3.2 is not included in this version (but only in the full version)

- lines 397-400: is it correct that all of your existential results yield inefficient algorithms? Then why only mention the efficient computation of 1/4-MMS as an open problem, and not the other ones?




Typos:

- line 83: world -> worlds
- line 100: max -> min
- line 101: finds -> find
- line 118: max -> min
- line 165: am -> an
- lines 441-447: double citation

---

> ### Author Rebuttal · Authors · 2023-08-03
>
> We appreciate your feedback. We will enhance the writing accordingly for the final version.
>
> Regarding the first question,
> - for Additive valuation, the state-of-the-art gives a constructive algorithm which is not polynomial but yields a PTAS.
> - In Submodular valuation class, a 10/27-MMS allocation can be computed in polynomial time.
> - For Fractionally Subadditive valuation, there is a polynomial algorithm that achieves 1/8 approximation. But the result on the best - approximation factor 0.219225 is existential.
> - In Subadditive valuation, the result on the best approximation factor is existential.
>
> Regarding your second question, to achieve a better approximation with the same technique, one idea would be to allocate more items to agents in the first phase. We believe it promising that this extension gives a better approximation factor; however, the analysis could not be trivially extended and new ideas will be needed for the proof of correctness. The more fundamental obstacle is that using this approach, no matter how many items we allocate in the first phase, the approximation factor converges to 1/4. Thus, in order to improve the approximation factor beyond 1/4, new techniques are required.

---

> > ### Comment · Reviewer_Y5Uh · 2023-08-13
> >
> > Thank you for your detailed response. I think adding a brief discussion in the paper about these two points will be helpful for the readers.

---

### Author Rebuttal · Authors · 2023-08-03

We found all the comments fair and valid. Therefore, we would like to use this opportunity to thank all the reviewers for the time and effort they dedicated to our paper. We will apply all the comments in the final version.

---

### Decision · Program_Chairs · 2023-09-21

**Decision:**

Accept (poster)

**Comment:**

I am recommending this paper for acceptance. Overall the reviewers were unanimously supportive on this paper, and that it makes a clear theoretical contribution worthy of being presented at NeurIPS (with scores of 3x7s and 1x6). The rebuttal helped clarify some smaller technical concerns.